# VOVTRACK: EXPLORING THE POTENTIALITY IN VIDEOS FOR OPEN-VOCABULARY OBJECT TRACKING

## ABSTRACT

Open-vocabulary multi-object tracking (OVMOT) represents a critical new challenge involving the detection and tracking of diverse object categories in videos, encompassing both seen categories (base classes) and unseen categories (novel classes). This issue amalgamates the complexities of open-vocabulary object detection (OVD) and multi-object tracking (MOT). Existing approaches to OVMOT often merge OVD and MOT methodologies as separate modules, predominantly focusing on the problem through an image-centric lens. In this paper, we propose VOVTrack, a novel method that integrates object states relevant to MOT and video-centric training to address this challenge from a video object tracking standpoint. First, we consider the tracking-related state of the objects during tracking and propose a new prompt-guided attention mechanism for more accurate localization and classification (detection) of the time-varying objects. Subsequently, we leverage raw video data without annotations for training by formulating a self-supervised object similarity learning technique to facilitate temporal object association (tracking). Experimental results underscore that VOVTrack outperforms existing methods, establishing itself as a state-of-the-art solution for open-vocabulary tracking task. We have included the source code in the supplementary material.

## 1 INTRODUCTION

Multi-Object Tracking (MOT) is a fundamental task in computer vision and artificial intelligence, which is widely used for video surveillance, media understanding, *etc*. In the past years, plenty of datasets, *e.g.*, MOT-20 (Dendorfer et al., 2020), DanceTrack (Sun et al., 2022), KITTI (Geiger et al., 2012), as well as the algorithms, *e.g.*, SORT (Wojke et al., 2017), Tracktor (Sridhar et al., 2019), FairMOT (Zhang et al., 2021), have been proposed for MOT problem. However, most previous works focus on the tracking of simple object categories, *i.e.*, humans and vehicles. Actually, it is important for the perception of various categories of objects in many real-world applications. Some recent works have begun to study the tracking of generic objects. TAO (Dave et al., 2020) is the first large dataset for the generic MOT, which includes 2,907 videos and 833 object categories. The later GMOT-40 (Bai et al., 2021) includes 10 categories and 40 videos with dense objects in each video.

With the development of Artificial General Intelligence (AGI) and multi-modal foundation models, open-world object perception has become a popular topic. Open-vocabulary object detection (OVD) is a new and promising task because of its generic settings. It aims to identify the various categories of objects from an image, including both the categories that have been seen during training (namely base classes) and not seen (namely novel classes). Although OVD has been studied in a series of works (Dhamija et al., 2020; Joseph et al., 2021; Doan et al., 2024), the literature on open-vocabulary (multi-) object tracking is rare. The nearly sole work (Li et al., 2023) builds a benchmark for open-vocabulary multi-object tracking (OVMOT) based on TAO (Dave et al., 2020). The authors also develop a simple framework for this problem consisting of a detection head and a tracking head. The detection head *is directly taken from an existing OVD algorithm*, *i.e.*, DetPro (Du et al., 2022), which is used to detect the open-vocabulary categories of objects in each frame without considering any tracking-related factors. Then the tracking head is used to learn the similarities among the detected objects in different frames. Since the lack of video data with open-vocabulary tracking annotations, the approach in Li et al. (2023) uses a data hallucination strategy to generate the image pairs for training the tracking head. However, the generated image pairs ignore the adjacent

continuity and temporal variability of the objects in a video sequence. As discussed above, the existing method for open-vocabulary tracking simply combines the approaches of OVD and MOT in series as independent modules. It does not consider the object states during tracking, *e.g.*, mutual occlusion, motion blur, *etc.*, and does not make use of the sequential information in the videos.

Therefore, in this work, we aim to handle the open-vocabulary object tracking from the standpoint of continuous videos. The comparison between our method and that in Li et al. (2023) can be intuitively seen from Figure 1. Specifically, we first consider the various states of the objects during tracking. Our basic idea is the damaged objects (*e.g.*, with occlusion, motion blur, *etc.*) should be weakened for foreground object feature learning. This way, we model these states as the prompts, which are used to calculate the attention weights of each generated detection proposal during training the object detection network. This methodology can provide more accurate detection (localization and classification) results of the time-varying objects during tracking. Second, to fully utilize the raw videos without open-vocabulary track-

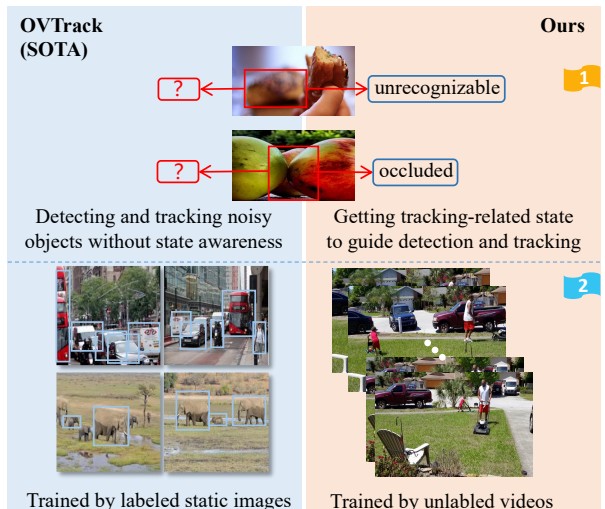

Figure 1: Comparison between prior (Li et al., 2023) and our methods.

ing annotations, we formulate the temporal association (tracking) task as a constraint optimization problem. The basic idea is that each object should maintain consistency across different frames, allowing us to leverage object consistency to effectively learn the object appearance similarity features for temporal association. Specifically, we formulate the object appearance self-consistency as the *intra-consistency*, and the spatial-appearance mutual-consistency as the *inter-consistency* learning problem, in which we also consider the object category consistency. These consistencies work together to learn the object appearance similarity features for the temporal association (tracking) sub-task, thus enhancing the overall performance of OVMOT. Notably, the existing OVMOT dataset TAO provides 534.1K (frames) of unlabeled video data. Compared to the 18.1K annotated samples, the unlabeled ones have a huge potentiality to be explored for training. Our self-supervised training method obtains significant performance improvements only using the raw data thus effectively alleviating the burden of annotations for OVMOT. The main contributions of this work are:

- We propose a new tracking-related prompt-guided attention for the localization and classification (detection) in the open-vocabulary tracking problem. This method takes notice of the states of the time-varying objects during tracking, which is different from the open-vocabulary object detection from a single image.
- We develop a self-supervised object similarity learning strategy for the temporal association (tracking) in the OVMOT problem. This strategy, for the first time, makes full use of the raw video data without annotation for OVMOT training, thus addressing the problem of training data shortage and eliminating the heavy burden of annotation of OVMOT.
- Experimental results on the public benchmark demonstrate that the proposed VOVTrack achieves the best performance with the same training dataset (annotations), and comparable performance with the methods using a large dataset (CC3M) for training.

## 2 RELATED WORK

**Multiple object tracking.** The prevailing paradigm in MOT is the tracking-by-detection framework (Andriluka et al., 2008), which involves detecting objects in each frame first, and then associating objects across different frames using various cues such as object appearance features (Bergmann et al., 2019; Fischer et al., 2023; Leal-Taixé et al., 2016; Milan et al., 2017; Pang et al., 2021; Sadeghian et al., 2017; Wojke et al., 2017; Cai et al., 2022), 2D motion features (Zhou et al., 2020;

Saleh et al., 2021; Xiao et al., 2018; Qin et al., 2023; Du et al., 2023), or 3D motion features (Huang et al., 2023; Luiten et al., 2020; Wang et al., 2023; Krejčí et al., 2024; Ošep et al., 2018; Sharma et al., 2018b). Some methods leverage graph neural networks (Bochinski et al., 2017; Ding et al., 2023) or transformers (Meinhardt et al., 2022; Sun et al., 2020; Zeng et al., 2022; Zhou et al., 2022d) to learn the association relationships between objects, thereby enhancing tracking performance. To broaden the object categories of the MOT task, the TAO benchmark (Dave et al., 2020) has been proposed for studying MOT under the long-tail distribution of object categories. On this benchmark, relevant methods include AOA (Du et al., 2021), GTR (Zhou et al., 2022d), TET (Li et al., 2022), QDTrack (Fischer et al., 2023), *etc*. While these methods perform well, they are still limited to pre-defined object categories, which makes them unsuitable for diverse open-world scenarios. Differently, this work handles OVMOT problem, which contains categories not seen during training.

**Open-world/vocabulary object detection.** Unlike traditional object detection with closed-set categories that appear at the training time. The task of open-world object detection aims to detect salient objects in an image without considering their specific categories (Dhamija et al., 2020; Joseph et al., 2021; Doan et al., 2024). This allows the method to detect objects of categories beyond those present in the training set. However, such methods do not classify objects into specific categories and only regard the object classification task as a clustering task (Joseph et al., 2021), achieving classification by calculating the similarity between objects and different class clusters. Consistent with the objective of open-world detection, open-vocabulary object detection requires the identification of categories not seen in the training set. However, unlike open-world detection, open-vocabulary object detection needs to predict the specific object categories (Zareian et al., 2021). To achieve this, some works (Bansal et al., 2018; Rahman et al., 2020) train the detector with text embeddings. Recently, pre-trained vision-language models ,*e.g.,* CLIP (Radford et al., 2021) connect visual concepts with textual descriptions, which has a strong open-vocabulary classification ability to classify an image with arbitrary categories described by language. Based on this, many works (Gu et al., 2022; Zhou et al., 2022c; Wu et al., 2023) utilize pre-trained vision-language models to achieve open-vocabulary object detection and few-shot object detection. In addition, some studies (Du et al., 2022; Zhou et al., 2022a;b) further enhance the effectiveness of prompt embeddings of class descriptions using prompt learning methods, thereby improving the results of open-vocabulary detection. Different from the above detection methods developed for single image, in this work, we propose a prompt-guided training method designed for open-vocabulary detection in continues videos, which can effectively enhance the detection performance for the video tracking task.

**Open-world/vocabulary object tracking.** There are relatively few works addressing the task of open-world tracking. Related approaches aim to segment or track all the moved objects in the video (Mitzel & Leibe, 2012; Dave et al., 2019) or handle the generic object tracking (Ošep et al., 2016; 2018; 2020) using a class-agnostic detector. Recently, the TAO-OW benchmark (Liu et al., 2022) is proposed to study open-world tracking problems, but its limitation lies in evaluating only class-agnostic tracking metrics without assessing class-related metrics. To make the setting more practical, OVTrack (Li et al., 2023) first brings the setting of open vocabulary into the tracking task, which also develops a baseline method and benchmark based on the TAO dataset. However, the method in Li et al. (2023) directly uses an existing OVD algorithm for detection, and its training process only utilizes static image pairs and ignores the information from video sequences. Differently, we consider the tracking-related object states for detection, and also propose a self-supervised video-based training method designed for open-vocabulary tracking, making full use of video-level information to enhance the performance of open-vocabulary tracking.

# 3 PROPOSED METHOD

## 3.1 OVERVIEW AND VOVTRACK FRAMEWORK

OVMOT requires localizing, tracking, and recognizing the objects in a video, whose problem formulation is provided in Appendix 1. We first describe the framework of our VOVTrack, which mainly includes the object localization, object classification, and temporal association modules, as shown in Figure 2. For improving the localization and classification, in Section 3.2, we design a tracking-state-aware prompt-guided attention mechanism, which can help the network learn more effective object detection features. For learning the temporal association similarity, in Section 3.3, we pro-

pose a video-based self-supervised method to train the association network, which considers the appearance intra-/inter-consistency and category consistency, to enhance the tracking performance.

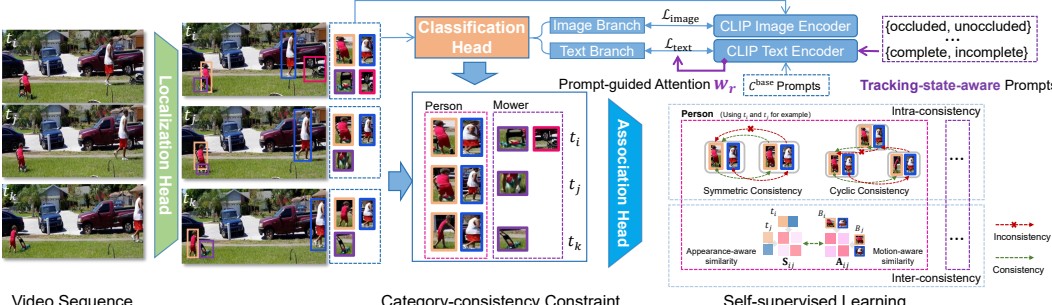

Figure 2: Training framework consists of three parts: first is the localization head used to localize objects of all categories in the video as region candidates; the second is the CLIP distilled classification head consisting of image and text branches, which uses tracking-sate-aware prompts to guide the model in focusing on object states while learning classification features, thereby better distinguishing the OV categories; and the third part is the association head that utilizes intra/inter-consistency between the same objects in different frames to learn association features in a self-supervised way.

**Localization**: We employ the class-agnostic object proposal generation approach in Faster R-CNN (Ren et al., 2015) to localize objects for both base and novel categories $\mathcal{C}$ in the video. As supported by prior researches (Dave et al., 2019; Gu et al., 2022; Zhou et al., 2022c; Li et al., 2023), the localization strategy has shown robust generalization capabilities towards the novel object class $\mathcal{C}^{\text{novel}}$. During the training phase, we leverage the above-mentioned generated RPN proposals as the initial object candidates $P$. The localization result of each candidate $r \in P$ is the bounding box $\mathbf{b} \in \mathbb{R}^4$. For each $\mathbf{b}$, we also obtain a confidence $p_c \in \mathbb{R}$ derived from the classification head. To refine the localization candidates, besides the classical non-maximum suppression (NMS), we also use this confidence $p_c$. For a more effective $p_c \in \mathbb{R}$ in the classification head, we use a prompt-guided attention strategy, which will be discussed in later *Section 3.2*.

**Classification**: Existing closed-set MOT trackers only track several specific categories of objects, which do not need to provide the class of each tracked object. This way, classification is a new sub-task of the OVMOT. To enable the framework to classify open-vocabulary object classes, following the OV detection algorithms (Gu et al., 2022; Zhou et al., 2022c; Wu et al., 2023), we leverage the knowledge from the pre-trained model, *i.e.*, CLIP (Radford et al., 2021), to help the network recognize objects belonging to the novel classes $\mathcal{C}^{\text{novel}}$. We distill the classification head using the CLIP model to empower the network's classification head to recognize new objects. Specifically, as shown in Figure 2, after obtaining the RoI feature embeddings $\mathbf{f}_r$ from the localization head, we design the classification head with a text branch and an image branch to generate embeddings $\mathbf{f}_r^{\text{text}}$ and $\mathbf{f}_r^{\text{img}}$ for each $\mathbf{f}_r$. The supervisions of these heads are generated by the CLIP text and image embeddings. We use the method in Du et al. (2022) for CLIP encoder pre-training.

First, we align the text branch with the CLIP text encoder. For $\forall c \in \mathcal{C}^{\text{base}}$, the class text embedding $\mathbf{t}_c$ of class $c$ can be generated by the CLIP text encoder $\mathcal{T}(\cdot)$ as $\mathbf{t}_c = \mathcal{T}(c)$. We compute the affinity between the predicted text embeddings $\mathbf{f}_r^{\text{text}}$ and their CLIP counterpart $\mathbf{t}_c$ as

$$z(r) = \left[ \cos\left(\mathbf{f}_r^{\text{text}}, \mathbf{t}_{\text{bg}}\right), \cos\left(\mathbf{f}_r^{\text{text}}, \mathbf{t}_1\right), \cdots, \cos\left(\mathbf{f}_r^{\text{text}}, \mathbf{t}_{|\mathcal{C}^{\text{base}}|}\right) \right],$$

$$\mathcal{L}_{\text{text}} = \frac{1}{|P|} \sum_{r \in P} w_r \mathrm{L}_{\text{CE}} \left( \text{softmax}(\frac{z(r)}{\lambda}), c_r \right), \tag{1}$$

where $\mathbf{t}_{\text{bg}}$ is a background prompt learned by treating the background candidates as a new class, $w_r$ is the tracking-related prompt-guided attention weight (*described in Section 3.2*), $\lambda$ is a temperature parameter, $\mathrm{L}_{\text{CE}}$ is the cross-entropy loss and $c_r$ is the class label of $r$.

Then, we align each image branch embedding $\mathbf{f}_r^{\text{img}}$ with the CLIP image encoder $\mathcal{I}(\cdot)$. For each candidate object $r$, we input the corresponding cropped image to $\mathcal{I}(\cdot)$ and get the CLIP image embedding $\mathbf{i}_r$. We minimize the distance between the corresponding $\mathbf{f}_r^{\text{img}}$ and $\mathbf{i}_r$ as

$$\mathcal{L}_{\text{image}} = \frac{1}{|P|} \sum_{r \in P} \left\| \mathbf{f}_r^{\text{img}} - \mathbf{i}_r \right\|. \tag{2}$$

In the testing stage, with the embeddings $\mathbf{f}_r^{\text{text}}$ and $\mathbf{f}_r^{\text{img}}$ derived from the text branch and image branch, respectively, we can obtain the corresponding classification probabilities $p_c^{\text{text}}$ and $p_c^{\text{img}}$ of each $r$ belonging to the class $c$, using the $\mathbf{z}(r)$ and softmax operation in Eq. (1). After that, we use the fusion strategy in Gu et al. (2022) to calculate the final classification probability $p_c$.

**Association**: The association head is to associate the detected objects with the same identification across frames, the main purpose of which is to learn the object features for similarity measurement.

In the training stage, given the detected objects from the localization head, we use the appearance embedding to extract the features for association. For training the appearance embedding model, a straightforward method is to select an object as the anchor, and its positive/negative samples for learning the object similarity. This method requires object identification (ID) annotations for positive/negative sample selection. However, as an emerging problem, OVMOT does not have enough available training video datasets with tracking ID labels. Previous work (Li et al., 2023) uses data augmentation to generate the image pairs for training the association head, which, however, ignores the temporal information in the videos. In this work, we propose to leverage the *unlabeled videos* to train the association network in a *self-supervised strategy*, *which will be discussed in Section 3.3*.

In the inference stage, we use appearance feature similarity to associate history tracks with the objects in the current frame. As in Li et al. (2023), we evaluate the similarity between history tracks and detected objects using both bi-directional softmax (Fischer et al., 2023) and cosine similarity metrics. Following the classical MOT approaches, if the similarity score exceeds a matching threshold, we assign the object to the track. If the object doesn't correspond to any existing track, a new track is initiated if its detection confidence score surpasses a threshold, otherwise, it is disregarded.

### 3.2 Tracking-State-Aware Prompt Guided Attention

**Tracking-state-aware prompt.** In the classification head of existing open-vocabulary detection methods, when selecting region candidates for calculating classification loss, they only consider whether the maximal IOU between the candidates and ground-truth box exceeds a threshold. For this tracking problem, we further consider whether the states of the candidates are appropriate for training the detection network.

As is well known, the objects present many specific states during tracking, such as occlusion/out-of-view/motion blur, *etc*, which are more frequent than the object detection in static images. So, it is important to identify such object states to achieve more accurate detection and tracking results. However, these states have often been overlooked in past methods because the labels of these states are difficult to obtain, not to mention incorporating them into the network training.

Prompt, as a burgeoning concept, can be used to bridge the gap between the vision and language data based on the cross-modal pre-trained models. We consider using specially designed prompts to perceive the tracking states of the objects and integrate such states into the model training. We refer to such prompts as 'tracking-state-aware prompts'. We specifically employ pairs of adjectives with opposite meanings to describe the object states during tracking. For example, 'unoccluded and occluded'. As shown in classification head of Figure 2, we add $M$ pairs of tracking-state-aware prompt pairs, denoted as $\{p_{\text{pos}}^m, p_{\text{neg}}^m\}_{m=1}^M$, into our framework model. Next, we present how to use these promotes during training.

**Prompt-guided attention.** To utilize these tracking-state-aware prompts guiding model training, we encode these prompts through the CLIP text encoder $\mathcal{T}(\cdot)$ to obtain state embeddings $\{\mathbf{t}_{\text{pos}}^m, \mathbf{t}_{\text{neg}}^m\}_{m=1}^M$, and calculate the prompt-guided attention weight $w_r$ as

$$\mathbf{z}(r, m) = \left[\cos\left(\mathbf{f}_r^{\text{text}}, \mathbf{t}_{\text{pos}}^m\right), \cos\left(\mathbf{f}_r^{\text{text}}, \mathbf{t}_{\text{neg}}^m\right)\right], \quad w_r = \frac{1}{|M|}\sum_{m=1}^M \left(\text{softmax}\,\frac{\mathbf{z}(r, m)}{\lambda}\right)_1, \quad (3)$$

where $\mathbf{f}_r^{\text{text}}$ is the text embedding of a region candidate $r$, and $()_1$ represents using the probability of the positive state as the attention for this pair. Note that, in our definition, the positive states always denote that the object state is beneficial to the object feature learning, *e.g.*, unoccluded, unobscured. From the above analysis, we know that the attention obtained from the tracking-state-aware prompts evaluates the various object states, resulting in prompt-guided attention values $w_r \subseteq [0, 1]$.

**Piecewise weighting strategy.** Next, we use this prompt-guided attention to help the model better utilize high-quality candidates for training and filter out the low-quality noisy candidates. Specifically, we divide the $w_r$ into three levels: low ($d_{\text{low}}$), medium and high ($d_{\text{high}}$). For embedding features $\mathbf{f}_r^{\text{text}}$ with $w_r < d_{\text{low}}$, we regard such features as low-quality state, and filter them out during training, by assigning $w_r$ to 0. For $d_{\text{low}} \leq w_r \leq d_{\text{high}}$, we regard that even if these features are not of high quality, they still contribute to training, and retain their original weights $w_r$. For $w_r > d_{\text{high}}$, we regard the features of these regions as particularly suitable ones for the model to learn tracking-related features, so we assign $w_r$ to 1 as an award. After that, we apply the re-assigned $w_r$ to Eq. (1), thereby integrating the object tracking-related state into the model training, which enables the network to better learn the object representations specifically for the tracking task.

## 3.3 SELF-SUPERVISED OBJECT ASSOCIATION WITH RAW VIDEO DATA

Considering the lack of annotated videos for OVMOT, we develop a self-supervised approach to train the association network by leveraging the consistency among the same objects during a video.

### 3.3.1 FORMULATION

**Objective function.** To learn the object appearance feature, we consider two aspects. The first one is **intra factor**, *i.e.*, the self-consistency of appearance for the same object at different times. The second one is **inter factor**, *i.e.*, the mutual-consistency between the appearance and motion cues during tracking. This way, we formulate the optimized objective function as

$$\max \quad S = S_{\text{intra}}(t_i, t_j) + \alpha \cdot S_{\text{inter}}(t_i, t_j), \quad s.t. \quad \forall t_i, t_j \in \mathcal{T}_c, \quad \forall c \in \mathcal{C}, \tag{4}$$

where $S$ represents the overall consistency objective to be maximized, $S_{\text{intra}}$ and $S_{\text{inter}}$ denote the intra and inter consistency measures respectively, while $\alpha$ is a weight to balance them. Besides, considering the diversity of object categories in the OVMOT problem, we add the object category consistency constraint for the consistency learning. Specifically, we construct the intervals, *e.g.*, $\mathcal{T}_c$, which contain several frames only with the same object category $c$, in which we select $t_i, t_j$. This is because we aim to learn the feature in a self-supervised manner without ID annotation, the objects with various categories may bring about interference.

**Long-short-interval sampling.** First, we consider the interval splitting of $\mathcal{T}_c$ in Eq. (4). We split the original videos into several segments of length $L$ and randomly sample the shorter sub-segments with various lengths from each segment. These short-term sub-segments are then concatenated to form the training sequence. Such training sequences include long-short-term intervals. Specifically, we select the adjacent frames from the same sub-segment, which allow the association head to learn the consistency objectives under minor object differences. We also select the long-interval video frames from different sub-segments, which allow the association head to learn the similarity and variation of objects under large differences.

**Category-consistency constraint.** Then we consider the category consistency constraint in $\mathcal{T}_c$. After obtaining the sampled training sequences as discussed above, we utilize the localization head to extract object bounding boxes from each frame in the sequence. Since we only use unlabeled raw videos for training, we cannot directly obtain the object categories. To address this issue, we employ a clustering approach to group the bounding boxes based on their classification features. Specifically, we utilize the classification head to obtain the category features for the candidate objects from all frames. Then we cluster the candidate objects' category features as different clusters. After clustering the candidate objects' category features into distinct groups, we can treat each cluster as a separate category. As illustrated in Figure 2, we proceed to conduct self-supervised learning on the objects that belong to the same category cluster and are sourced from different frames.

### 3.3.2 SELF-SUPERVISED LOSS CONSTRUCTION

We next model the consistency learning problem in Eq. (4) as a self-supervised learning task.

**Intra-consistency loss.** After getting the training samples (object bounding boxes in different frames of the sample training sequence within the category clustering), we first use the CNN network to extract the appearance feature from the association head for all objects in frame $t_i$ to construct the feature matrix $\mathbf{F}_i$. The main idea of the intra-consistency loss is to leverage the self-consistency of

the same objects at different times (frames). We utilize the following two types of similarity transfer relationships, *i.e.*, pair-wise symmetry and triple-wise cyclicity.

• *Consistent learning from the symmetry*: For a pair of frames $t_i$ and $t_j$ in the video, we can get the object similarity matrix between them as

$$\mathbf{M}_{ij} = \mathbf{F}_i \cdot (\mathbf{F}_j)^{\mathrm{T}}. \tag{5}$$

We then compute the *normalized similarity matrix* $\mathbf{S}_{ij} \in [0, 1]$ based on the above similarity matrix $\mathbf{M}_{ij}$ by temperature-adaptive row softmax as

$$\mathbf{S}(r, c) = f_{r,c}(\mathbf{M}) = \frac{\exp(\tau \mathbf{M}(r, c))}{\sum_{c=1}^{C} \exp(\tau \mathbf{M}(r, c))} \tag{6}$$

where $r, c$ represent the indices of row and column in $\mathbf{M}$, respectively. Here $C$ is the number of columns for $\mathbf{M}$, and $\tau$ is the adaptive temperature adjustable parameter.

The normalized similarity matrix $\mathbf{S}_{ij}$ can be regarded as a mapping (object association relation) from frame $t_i$ to frame $t_j$ ($\mathbf{S}_{ij} : t_i \rightarrow t_j$). In other words, we can select the maximum value of each row of $\mathbf{S}$ as the matched objects between frames $t_i$ and $t_j$. Similarly, we get the reversed mapping from $t_j$ to $t_i$ as $\mathbf{S}_{ji} : t_j \rightarrow t_i$. We calculate the pair-wise symmetric-similarity matrix as $\mathbf{E}_{\mathrm{pair}} = \mathbf{S}_{ij} \cdot \mathbf{S}_{ji}$, where $\mathbf{E}_{\mathrm{pair}}$ can be regarded as a symmetric mapping: $t_i \rightleftarrows t_j$, *i.e.*, from $t_i$ and return $t_i$. If the objects in frames $t_i$ and $t_j$ are the same, the result $\mathbf{E}_{\mathrm{pair}}$ should be an identity matrix, which can be used to construct the self-supervision loss. However, due to the object differences in different frames, this condition may not be always satisfied. Therefore, we need to supervise it deliberately, which will be discussed later.

• *Consistent learning from the cyclicity*: Besides the pair-wise symmetric similarity, we further consider the triple-wise circularly consistent similarity. Specifically, given the similarity matrix $\mathbf{S}_{ij}$ between two frames $t_i$ and $t_j$, as well as $\mathbf{S}_{jk}$ between frames $t_j$ and $t_k$, we aim to build the consistent similarity relation among this triplet, *i.e.*, frames $t_i, t_j$ and $t_k$. To do this, we first calculate the third-order similarity matrix as

$$\mathbf{M}_{ik} = \mathbf{M}_{ij} \cdot \mathbf{M}_{jk}, (i \neq j \neq k), \tag{7}$$

where $\mathbf{M}_{ik}$ represents the similarity between the objects in frames $t_i$ and $t_k$, through the frame $t_j$. We then compute the normalized similarity matrix using Eq. (6) as

$$\mathbf{S}_{ik} = f(\mathbf{M}_{ik}), \quad \mathbf{S}_{ki} = f((\mathbf{M}_{ik})^{\mathrm{T}}), \tag{8}$$

where $\mathbf{S}_{ik}$ represents the mapping $t_i \rightarrow t_j \rightarrow t_k$ while $\mathbf{S}_{ki}$ represents the mapping along $t_k \rightarrow t_j \rightarrow t_i$. Then, we calculate the transitive-similarity matrix: $\mathbf{E}_{\mathrm{trip}} = \mathbf{S}_{ik} \cdot \mathbf{S}_{ki}$, where $\mathbf{E}_{\mathrm{tri}}$ can be regarded as the mapping: $t_i \rightleftarrows t_j \rightleftarrows t_k$ (from $t_i$ and return $t_i$).

For convenience, we note the matrices $\mathbf{E}_{\mathrm{pair}}$ and $\mathbf{E}_{\mathrm{trip}}$ as $\mathbf{E}$, which should have the property that their diagonal elements are either 1 or 0, while all other elements are 0, in an ideal case. This means that the diagonal elements of $\mathbf{E}$ must be greater than or equal to the other elements. Based on this consideration, following Wang et al. (2020); Feng et al. (2024), we use the following loss $\mathrm{L}(\mathbf{E}) = \sum_r \mathrm{relu}(\max_{c \neq r} \mathbf{E}(r, c) - \mathbf{E}(r, r) + m)$, where $r, c$ denote the indices of row and column in $\mathbf{E}$. This loss denotes that we penalize the cases where the max non-diagonal element $\mathbf{E}(r, c)$ ($c \neq r$) in a row $r$, exceeds the corresponding diagonal elements $\mathbf{E}(r, r)$ with a margin $m$. The margin $m \geq 0$ is a parameter used to control the punishment scope between $\mathbf{E}(r, c)$ and $\mathbf{E}(r, r)$. This loss helps $\mathbf{E}$ approach the identity matrix while addressing cases where there are unmatched targets in a row through a margin $m$. Finally, we define our self-supervised consistency learning loss (intra) as $\mathcal{L}_{\mathrm{intra}} = \mathrm{L}(\mathbf{E}_{\mathrm{pair}}) + \mathrm{L}(\mathbf{E}_{\mathrm{trip}})$.

**Inter-consistency loss.** The inter-consistency loss makes use of the consistency between the spatial position continuity and the appearance similarity of the objects at different times. Given a bounding box list $B_i$ and $B_j$ of adjacent frames $i$ and $j$ in the video, we compute the Intersection over Union (IoU) matrix $\mathbf{M}_{ij}$ between bounding boxes $B_i$ and $B_j$, which quantifies the overlap between them. Based on a specified threshold $\mathrm{IoU}_{\mathrm{thres}}$, we create an assignment matrix $\mathbf{A}_{ij}$ such that

$$\mathbf{A}_{ij} = \begin{cases} 1 & \text{if } \mathbf{M}_{ij} > \mathrm{IoU}_{\mathrm{thres}} \\ 0 & \text{otherwise} \end{cases}, \tag{9}$$

This assignment matrix $\mathbf{A}_{ij}$ indicates whether pairs of objects are considered similar, facilitating the optimization of the spatial consistency objective. Furthermore, we utilize the previously obtained normalized similarity matrix $\mathbf{S}_{ij}$ to establish the inter-consistency loss, using the binary cross-entropy loss function $\mathrm{L}_{\mathrm{BCE}}$, as $\mathcal{L}_{\mathrm{inter}} = \mathrm{L}_{\mathrm{BCE}}(\mathbf{S}_{ij}, \mathbf{A}_{ij})$. This formulation allows us to measure the discrepancy between the normalized similarity matrix $\mathbf{S}_{ij}$ and the assignment matrix $\mathbf{A}_{ij}$, thereby enforcing spatial consistency across the detected objects in frames $t_i$ and $t_j$.

Finally, we have transformed the optimization problem in Eq. (4) into a self-supervised loss as $\mathcal{L} = \mathcal{L}_{\mathrm{intra}} + \alpha \cdot \mathcal{L}_{\mathrm{inter}}$, where $\alpha$ is a weighting coefficient. This formulation incorporates both the intra- and inter-consistencies, as well as category consistency, among the frames with different intervals, thereby effectively exploring the potentiality of videos to enhance the association for OVMOT.

### 3.4 IMPLEMENTATION DETAILS

In Section 3.1, we use ResNet50 with FPN for localizing candidate regions. We set $\lambda$ in Eq. (1) as 0.007. In Section 3.2, we select four pairs of typical tracking state aware prompts, *i.e.*, 'complete and incomplete', 'unoccluded and occluded', 'unobscured and obscured', and 'recognizable and unrecognizable'. We set $d_{\mathrm{low}}$ as 0.3, $d_{\mathrm{high}}$ as 0.6. In Section 3.3, the segment length $L$ is set as 24. We use the clustering algorithm of K-means. We set the margin $m$ as 0.5 and the $\mathrm{IoU}_{\mathrm{thres}}$ as 0.9 . For $N$ frames in each sampled video sequence, we select $C_N^2$ and $C_N^3$ groups of frames to calculate the matrices in Eqs. (5) and (7). We also select $C_N^2$ groups of frames to calculate the inter-consistency loss. The weighting coefficients $\alpha$ is 0.9.

In the training stage, we first train the two-stage detector on the base classes of LVIS dataset (Gupta et al., 2019) referenced from Du et al. (2022), for 20 epochs, and use prompt-guided attention proposed in Section 3.2 to fine-tune the model's classification head for 6 epochs. Then we train the association head using static image pairs generated from Li et al. (2023) for 6 epochs and self-supervise the association head with TAO training dataset (Dave et al., 2020) without annotation for 14 epochs. In the inference stage, we select object candidates $P$ by NMS with IoU threshold 0.5. Additionally, we set the similarity score threshold as 0.35 and maintain a track memory of 10 frames.

## 4 EXPERIMENTS

### 4.1 DATASET AND METRICS

We follow Li et al. (2023) to select the dataset and metrics for evaluation. We leverage the comprehensive and extensive vocabulary MOT dataset TAO (Dave et al., 2020) as our benchmark for OVMOT. TAO is structured similarly to the LVIS (Gupta et al., 2019) taxonomy, categorizing object classes based on their frequency of appearance into frequent, common, and rare groups. We use the rare classes defined in LVIS as $\mathcal{C}^{\mathrm{novel}}$ and others as $\mathcal{C}^{\mathrm{base}}$. We evaluate the performance with the comprehensive metric tracking-every-thing accuracy (TETA), which consists of the accuracies of localization (LocA), classification (ClsA), and association (AssocA).

### 4.2 COMPARATIVE RESULTS

We compare our method with the latest trackers, TETer (Li et al., 2022) and QDTrack (Fischer et al., 2023), which are trained on both $\mathcal{C}^{\mathrm{base}}$ and $\mathcal{C}^{\mathrm{novel}}$. We include the classical trackers like DeepSORT (Wojke et al., 2017) and Tracktor++ (Bergmann et al., 2019) trained only on $\mathcal{C}^{\mathrm{base}}$ and enhanced by OVD method ViLD (Gu et al., 2022) to achieve open-vocabulary tracking, as baselines. We also compare our method with the state-of-the-art OVMOT method, OVTrack (Li et al., 2023). Besides, following Li et al. (2023), we compare with the existing trackers (DeepSORT, Tracktor++, OVTrack) equipped with the powerful OVD method, *i.e.*, RegionCLIP (Zhong et al., 2022) trained on the extensive CC3M (Sharma et al., 2018a) dataset.

As shown in Table 1, we present the OVMOT evaluation results on the TAO validation and test sets, divided into base classes $\mathcal{C}^{\mathrm{base}}$ and novel classes $\mathcal{C}^{\mathrm{novel}}$. We can see that our method outperforms all closed-set and open-vocabulary methods on both the validation and test sets. Even though QD-Track (Fischer et al., 2023) and TETer (Li et al., 2022) have seen novel classes during training on TAO, our method significantly outperforms them in all metrics on both base and novel classes.

Table 1: Result comparison. We evaluate our method against closed-set and open-vocabulary trackers on TAO validation and test sets. Here '✓' denotes using the corresponding dataset with annotations, and '†' represents only using the raw video without any annotation.

| Method | Classes | | Data | | | Base | | | | Novel | | | |
|---|---|---|---|---|---|---|---|---|---|---|---|---|---|
| | Base | Novel | CC3M | LVIS | TAO | TETA | LocA | AssocA | ClsA | TETA | LocA | AssocA | ClsA |
| **Validation set** | Base | Novel | CC3M | LVIS | TAO | TETA | LocA | AssocA | ClsA | TETA | LocA | AssocA | ClsA |
| QDTrack (Fischer et al., 2023) | ✓ | ✓ | - | ✓ | ✓ | 27.1 | 45.6 | 24.7 | 11.0 | 22.5 | 42.7 | 24.4 | 0.4 |
| TETer (Li et al., 2022) | ✓ | ✓ | - | ✓ | ✓ | 30.3 | 47.4 | 31.6 | 12.1 | 25.7 | 45.9 | 31.1 | 0.2 |
| DeepSORT (ViLD) (Wojke et al., 2017) | ✓ | - | - | ✓ | ✓ | 26.9 | 47.1 | 15.8 | 17.7 | 21.1 | 46.4 | 14.7 | 2.3 |
| Tracktor++ (ViLD) (Bergmann et al., 2019) | ✓ | - | - | ✓ | ✓ | 28.3 | 47.4 | 20.5 | 17.0 | 22.7 | 46.7 | 19.3 | 2.2 |
| DeepSORT + RegionCLIP* | ✓ | - | ✓ | ✓ | ✓ | 28.4 | 52.5 | 15.6 | 17.0 | 24.5 | 49.2 | 15.3 | 9.0 |
| Tracktor++ + RegionCLIP* | ✓ | - | ✓ | ✓ | ✓ | 29.6 | 52.4 | 19.6 | 16.9 | 25.7 | 50.1 | 18.9 | 8.1 |
| OVTrack (Li et al., 2023) | ✓ | - | - | ✓ | - | 35.5 | 49.3 | 36.9 | **20.2** | 27.8 | 48.8 | 33.6 | 1.5 |
| OVTrack + RegionCLIP* | ✓ | - | ✓ | ✓ | - | 36.3 | 53.9 | 36.3 | 18.7 | 32.0 | 51.4 | 33.2 | **11.4** |
| VOVTrack (Ours) | ✓ | - | - | ✓ | † | **38.1** | **58.1** | **38.8** | 17.5 | **34.4** | **57.9** | **39.2** | 6.0 |
| **Test set** | Base | Novel | CC3M | LVIS | TAO | TETA | LocA | AssocA | ClsA | TETA | LocA | AssocA | ClsA |
| QDTrack (Fischer et al., 2023) | ✓ | ✓ | - | ✓ | ✓ | 25.8 | 43.2 | 23.5 | 10.6 | 20.2 | 39.7 | 20.9 | 0.2 |
| TETer (Li et al., 2022) | ✓ | ✓ | - | ✓ | ✓ | 29.2 | 44.0 | 30.4 | 10.7 | 21.7 | 39.1 | 25.9 | 0.0 |
| DeepSORT (ViLD) (Wojke et al., 2017) | ✓ | - | - | ✓ | ✓ | 24.5 | 43.8 | 14.6 | 15.2 | 17.2 | 38.4 | 11.6 | 1.7 |
| Tracktor++ (ViLD) (Bergmann et al., 2019) | ✓ | - | - | ✓ | ✓ | 26.0 | 44.1 | 19.0 | 14.8 | 18.0 | 39.0 | 13.4 | 1.7 |
| DeepSORT + RegionCLIP* | ✓ | - | ✓ | ✓ | ✓ | 27.0 | 49.8 | 15.1 | 16.1 | 18.7 | 41.8 | 9.1 | 5.2 |
| Tracktor++ + RegionCLIP* | ✓ | - | ✓ | ✓ | ✓ | 28.0 | 49.4 | 18.8 | 15.7 | 20.0 | 42.4 | 12.0 | 5.7 |
| OVTrack (Li et al., 2023) | ✓ | - | - | ✓ | - | 32.6 | 45.6 | 35.4 | 16.9 | 24.1 | 41.8 | 28.7 | 1.8 |
| OVTrack + RegionCLIP* | ✓ | - | ✓ | ✓ | - | 34.8 | 51.1 | 36.1 | **17.3** | 25.7 | 44.8 | 26.2 | **6.1** |
| VOVTrack (Ours) | ✓ | - | - | ✓ | † | **37.0** | **56.1** | **39.3** | 15.5 | **29.4** | **52.4** | **31.2** | 4.5 |

*Note that, except the RegionCLIP (Zhong et al., 2022), all other methods (including 'Ours') use ResNet50 as backbone.

Additionally, DeepSORT and Tracktor++ with the open-vocabulary detector ViLD (Gu et al., 2022) are also trained in a supervised manner on TAO, while our method, trained in a self-supervised manner on TAO, surpasses them by a large margin. Although the results of OVTrack, the most competitive method, are better than other comparison methods, our method outperforms it in almost all metrics significantly on both base and novel classes.

Particularly, compared to our baseline method OVTrack, our method achieves improvements of 2.6% and 6.6% in base and novel TETA, respectively, and a 4.5% increase in novel ClsA. In the test set, base and novel TETA also show improvements of 4.4% and 5.3%, respectively. It is worth noting that even though RegionCLIP-related methods use an additional 3 million image data in CC3M for training, our method outperforms them in almost all metrics, with only ClsA slightly lower. This demonstrates the effectiveness of the proposed approach, which is very promising for the OVMOT task. We provide more analysis from the standpoint of data quantity for training in Appendix 2.

### 4.3 ABLATION STUDY

Table 2: Ablation studies on modules of prompt-guided attention and self-supervised association.

| Module | Ablation Methods | Base | | | | Novel | | | |
|---|---|---|---|---|---|---|---|---|---|
| | | TETA | LocA | AssocA | ClsA | TETA | LocA | AssocA | ClsA |
| Prompt-guided attention | w/o prompt-guided attention | 35.7 | 52.7 | 37.2 | 17.3 | 29.8 | 52.8 | 34.9 | 1.7 |
| | w/o piecewise weight strategy | 36.3 | 53.9 | 37.5 | 17.4 | 31.7 | 53.8 | 36.8 | 4.5 |
| Self-supervised association | w/o self-supervised learning | 36.3 | 55.5 | 36.4 | 17.1 | 31.3 | 55.1 | 34.7 | 4.0 |
| | w/o short-long-sampling | 37.3 | 57.2 | 36.9 | **17.7** | 33.1 | 56.9 | 37.2 | 5.1 |
| | w/o category consistency | 37.1 | 57.6 | 37.4 | 16.3 | 32.2 | 56.0 | 37.0 | 3.7 |
| | w/o intra-consistency | 37.0 | 57.0 | 36.7 | 17.4 | 32.3 | 56.1 | 36.0 | 4.9 |
| | w/o inter-consistency | 37.2 | 56.8 | 37.2 | 17.6 | 33.0 | 56.6 | 36.8 | 5.5 |
| | VOVTrack (Ours) | **38.1** | **58.1** | **38.8** | 17.5 | **34.4** | **57.9** | **39.2** | **6.0** |

In this section, we conduct the ablation studies on all components proposed in our method, including the ablation of prompt-guided attention, and the self-supervised learning related modules as:
- w/o prompt-guided attention ($w_r$): Removing the prompt-guided attention in Section 3.2.
- w/o piecewise weight strategy ($d_{low}$ and $d_{high}$): Removing the piecewise weighting strategy proposed in Section 3.2, by directly using the $w_r$ calculated by Eq. (3).
- w/o self-supervised learning: Removing the whole self-supervised learning strategy in Section 3.3.
- w/o short-long-sampling: Removing the short-long-interval sampling strategy in Section 3.3.
- w/o category consistency: Removing the category-aware object clustering in Section 3.3.
- w/o intra-consistency: Removing the intra-consistency consistency loss.
- w/o inter-consistency: Removing the inter-consistency loss in Section 3.3.

**Effectiveness of the state-aware prompt guided attention.** As shown in the first unit of Table 2, we can see that using prompt-guided attention as a weight coefficient during the training stage can effectively improve all metrics for both base and novel classes. The piecewise weighting strategy is also very effective, especially in improving the classification accuracy of novel classes.

**Effectiveness of the self-supervised consistent learning.** As shown in the second unit of Table 2, we can see that using self-supervised loss can effectively improve all metrics for both base and novel classes. Either the intra-consistency or the inter-consistency for appearance learning is effective for the association task, *i.e.*, 'AssocA'. Also, the interval sampling strategy allows samples to have a more diverse range of long and short cycles, improving the association-related metric. The category clustering strategy tries to gather the objects with the same category in a cluster, which is also helpful. To our surprise, the above strategies, in most cases, also effectively help improve classification ('ClsA') and localization ('LocA') accuracies. This is because the better association results can indirectly help to other two sub-tasks in OVMOT. We provide the discussion and analysis of the complementarity among different tasks in Appendix 3.

### 4.4 QUALITATIVE ANALYSIS

We conduct some qualitative analysis to more intuitively show the effect of our prompt-guided attention and the visualized comparison results of our method with the state-of-the-art algorithm.

**Illustrations of the proposed prompt-guided attention.** Figure 3 (a) shows cases of high prompt-guided attention, where we can see that the regions often have very distinctive category features, with no occlusion, and the image quality of the region is very high. In contrast, Figure 3 (b) presents cases of low prompt-guided attention, where we can observe that these regions often have issues such as heavy blurriness, occlusion, unclear visibility, and difficulty in identification. Such samples are very unsuitable for training the object localization and classification features, which are appropriately weakened through state-aware prompt-guided attention.

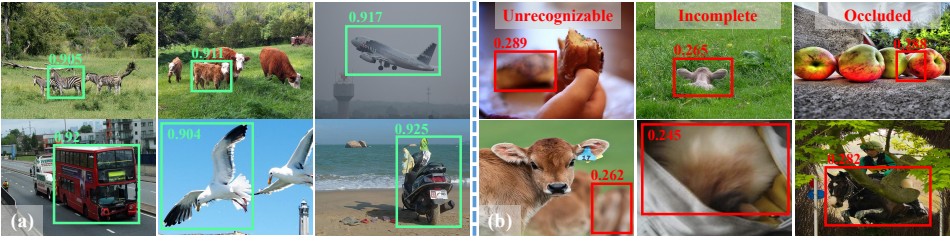

Figure 3: Illustration of regions with high (a) or low (b) prompt-guided attention, respectively.

**Comparison result visualization.** We show several visualization results in Figure 4. We can see that the proposed method provides better results than OVTrack (Li et al., 2023). In the first case of Figure 4 (a), our method provides an accurate object localization result and identifies the correct category. In the second case of Figure 4 (b), the tracking of a drone provided by the comparison method is wrong (different box colors denote different tracking IDs), also the classification is not correct. Our method can track it continuously. We also show some failure cases in Appendix 4.

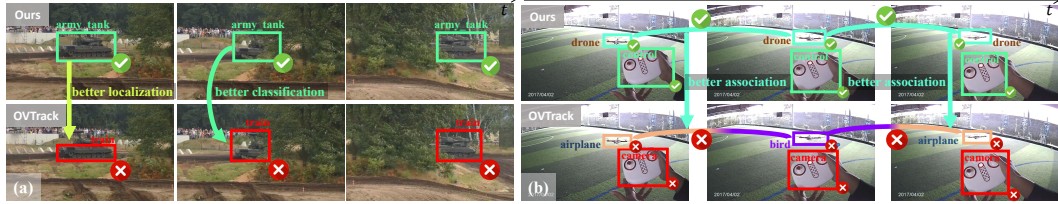

Figure 4: Compared OVMOT results of ours and OVTrack on some cases with novel classes.

## 5 CONCLUSION

In this work, we have developed a new method namely VOVTrack to handle the OVMOT problem from the perspective of video object tracking. For this purpose, we first consider the object state during tracking and propose tracking-state-aware prompt-guided attention, which improves the accuracy of object localization and classification (detection). Second, we develop an object similarity learning strategy for the temporal association (tracking) using only the raw video data without annotation, which unveils the power of self-supervised learning for open-vocabulary tracking tasks. Experimental results demonstrate the effectiveness of the proposed method and each component for open-vocabulary tracking.

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

# A  Appendix

## Appendix 1. OVMOT Problem

OVMOT requires the tracker to be capable of tracking objects from the open-vocabulary categories of objects. We first present the problem formulation of this task from the training and testing stages.

At training stage, the training data is $\left\{ \mathbf{X}^{\text{train}}, \mathcal{A}^{\text{train}} \right\}$ that contains video sequences $\mathbf{X}^{\text{train}}$ and their respective annotations $\mathcal{A}^{\text{train}}$ of the objects. Given one frame in the video, each annotation $\alpha \in \mathcal{A}^{\text{train}}$ consists of a 2D bonding box $\mathbf{b} = [x, y, w, h]$, a unified ID $d$ over the whole video, and a category label $c$, where $(x, y)$ is the center pixel coordinates and $(w, h)$ is the width and height of the box, the category belongs to the *base class* set, *i.e.*, $c \in \mathcal{C}^{\text{base}}$.

At the testing stage, the inputs consist of video sequences $\mathbf{X}^{\text{test}}$ and the set of all object classes $\mathcal{C} = \mathcal{C}^{\text{base}} \cup \mathcal{C}^{\text{novel}}$, where $\mathcal{C}^{\text{novel}}$ denotes the novel categories not appearing in the training set, *i.e.*, $\mathcal{C}^{\text{novel}} \cap \mathcal{C}^{\text{base}} = \emptyset$. OVMOT aims to obtain the trajectories of all objects in $\mathbf{X}^{\text{test}}$ belonging to classes $\mathcal{C}$. Each trajectory $\tau$ consists of a series of tracked objects $\tau_t$ at frame $t$, and each $\tau_t$ is composed of a 2D bounding box $\mathbf{b}$, and its object category $c$. Note that, during the testing stage, we need to evaluate not only the results on the base class $\mathcal{C}^{\text{base}}$, but also on the novel class $\mathcal{C}^{\text{novel}}$. The results on $\mathcal{C}^{\text{novel}}$ can validate the tracker's capability when facing objects from the open-vocabulary categories.

## Appendix 2. Training Data Analysis

As discussed above, we use the training dataset in TAO for association module training. Next, we will analyze our experimental results from the perspective of the data quantity used for training.

**TAO dataset.** As shown in the first row of Table 3, we can see that the original TAO dataset has very few annotated frames, with only 18.1k frames, and limited box annotations of 54.7k. This is because the annotations in TAO were made at 1 FPS, resulting in a very limited number of supervised frames and available annotations for training a robust tracker.

As shown in the next row, in our self-supervised method, we use all the raw video frames without requiring any annotations. We can see that the usable frame quantity has increased to 30 times compared to the original training set (with annotations). Also, the quantity of available object bounding boxes for self-supervised training has reached 399.9k, which is 7.5 times the original number of annotated ones. Moreover, by integrating the long-short-term sampling strategies, we can fully utilize all the long-short-term frames within in the TAO raw videos through our self-supervised method, thereby achieving better results.

Table 3: The number of frames and annotations can be used to train in LVIS, annotated TAO, TAO in our self-supervised paradigm and CC3M.

| Datasets | Frames | | Annotations (detections) | |
|---|---|---|---|---|
| TAO (Original training set) | 18.1k | | 54.7k | |
| TAO (Our self-supervision) | 534.1k | | 399.9k | |
| Datasets | Frames | | Annotations (detections) | |
| LVIS | base | novel | base | novel |
| | 99.3k | 1.5k | 1264.9k | 5.3k |
| Datasets | Frames | | Annotations (captions) | |
| CC3M | 3318.3k | | 3318.3k | |

We further discuss the results using the training datasets of LVIS and CC3M.

**LVIS dataset.** As shown in Table 1 in the main paper, the comparison methods QDTrack (Fischer et al., 2023) and TETer (Li et al., 2022) trained on the LVIS dataset with both base and novel classes, still yield poor results in TAO validation and test sets. This may be due to the imbalance in the data quantity of base and novel categories. Specifically, as seen in Table 3, although the LVIS dataset has a large number of frames and annotations for its base classes, the data for its novel classes is very limited, with the number of frames being $\frac{1}{66}$ and the number of annotations even less, at $\frac{1}{239}$.

**CC3M dataset.** We also list the data quantity of CC3M (Sharma et al., 2018a) in the last row of Table 3 to explain why our classification accuracy is slightly lower than the methods trained CC3M. We can see that the CC3M dataset is significantly larger, nearly 33 times the size of LVIS and 184 times that of TAO. In it, each frame caption also provides an average of about 10 words for training. The scenes and categories in the CC3M dataset are far more diverse than those in LVIS and TAO, which enables it to encounter a wider range of categories and achieve higher classification accuracy. However, it is noteworthy that, despite this, our method surpasses the results of the methods using CC3M in most metrics except classification, effectively demonstrating the effectiveness of our method.

APPENDIX 3. MODULE COMPLEMENTARITY ANALYSIS

When designing the entire framework, we also consider the complementarity of the localization, association, and association modules, enabling them to assist each other.

**Improving classification via association.** Following the baseline (Li et al., 2023), we use the most frequently occurring category within a trajectory as the category for all objects in that trajectory. This approach indirectly improves classification results through better associations. Such assistance explains the reason that category clustering operations in our self-supervised object association training effectively increase classification accuracy, as shown in Table 2 in the main paper.

**Improving localization via association.** Additionally, during the association process, some candidates from the localization module with low detection confidence scores are retained because their association similarity surpasses the threshold. This association similarity priority strategy ensures that valid targets are retained, which improves the accuracy of localization.

Similarly, better localization and classification results also help achieve improved association results, making our entire framework a cohesive entirety with multiple modules working collaboratively.

APPENDIX 4. FAILURE CASE ANALYSIS

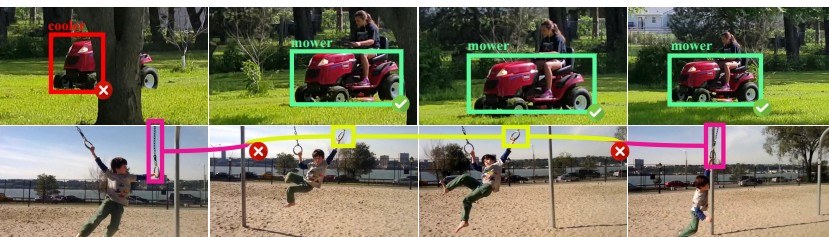

Figure 5: Failure case illustration.

We provide some failure cases in Figure 5. The first case illustrates a classification mistake due to significant occlusion. The second case shows the tracking errors caused by the distraction of object similarity and variability. We find that the OVMOT combined with the localization, classification, and tracking tasks has a significant challenge, yet it holds large research potential.

APPENDIX 5. MORE DETAILS IN THE PROPOSED METHOD

**Details during training.** As mentioned in the main paper regarding the experimental procedure, compared to using the existing Open-Vocabulary Detection (OVD) method (Du et al., 2022) directly for localization and classification in OVTrack (Li et al., 2023), we train the OVD process using the base classes of the LVIS dataset and incorporate tracking-related states into the training process (Section 3.2). This significantly enhanced the localization and classification results in open-vocabulary object tracking.

Additionally, in the training of the association module, different from our baseline method (Li et al., 2023) using the generated image pairs constructed by LVIS, we further introduce a self-supervised method for object similarity learning (Section 3.3). Specifically, we utilize all the video frames in the TAO (Dave et al., 2020) training dataset for self-supervised training, which makes full use of the consistency among the objects in a video sequence and greatly improves the association task results.

**Long-Short-Interval Sampling Strategy.** We consider the interval splitting of $\mathcal{T}_c$ in Eq. (4). As shown in Figure 6, we split the original videos into several segments of length $L$ and randomly sample the shorter sub-segments with various lengths from each segment. These short-term sub-segments are then concatenated to form the training sequence. Such training sequences include long-short-term intervals. Specifically, we select the adjacent frames from the same sub-segment, which allow the association head to learn the consistency objectives under minor object differences. We also select the long-interval video frames from different sub-segments, which allow the association head to learn the similarity and variation of objects under large differences.

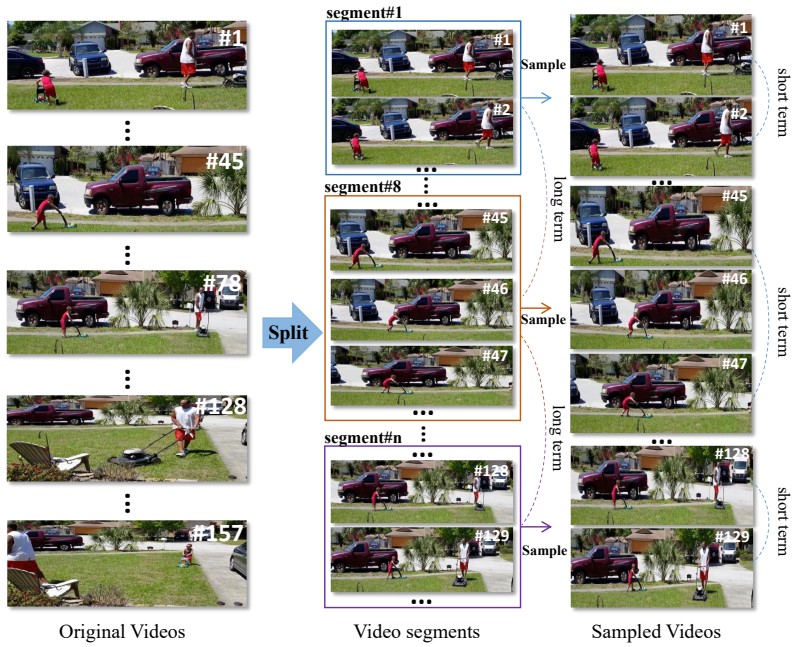

Figure 6: An illustration of interval sampling strategy.

**Metrics.** First, the localization accuracy (LocA) is determined through the alignment of all labeled boxes $\alpha$ with the predicted boxes of $\mathcal{T}$ without considering classification: $\text{LocA} = \frac{|\text{TPL}|}{|\text{TPL}|+|\text{FPL}|+|\text{FNL}|}$. Next, classification accuracy (ClsA) is calculated using all accurately localized TPL instances, by comparing the predicted semantic classes with the corresponding ground truth classes $\text{ClsA} = \frac{|\text{TPC}|}{|\text{TPC}|+|\text{FPC}|+|\text{FNC}|}$. Finally, association accuracy (AssocA) is determined using a comparable approach, by matching the identities of associated ground truth instances with accurately localized predictions $\text{AssocA} = \frac{1}{|\text{TPL}|}\sum_{b\in\text{TPL}}\frac{|\text{TPA}(b)|}{|\text{TPA}(b)|+|\text{FPA}(b)|+|\text{FNA}(b)|}$. The TETA score is computed as the mean value of the above three scores $\text{TETA} = \frac{\text{LocA}+\text{ClsA}+\text{AssocA}}{3}$.

APPENDIX 6. MORE VISUALIZATION CASES OF THE PROPOSED PROMPT-GUIDED ATTENTION.

To demonstrate the effectiveness of prompt-guided attention in target state perception and illustrate the necessity of filtering out low-quality objects, we present additional examples of low prompt-guided attention in Figure 7. The targets shown in the figure exhibit severe occlusion, incompleteness, or poor recognizability, which aligns with our initial design considerations for the prompts. These damaged targets can lead to network training issues where learning ambiguous target features limits the network's Open-Vocabulary (OV) generalization capability. Our proposed prompt-guided attention mechanism effectively suppresses this critical issue in OV settings, thereby significantly enhancing the perception of novel targets.

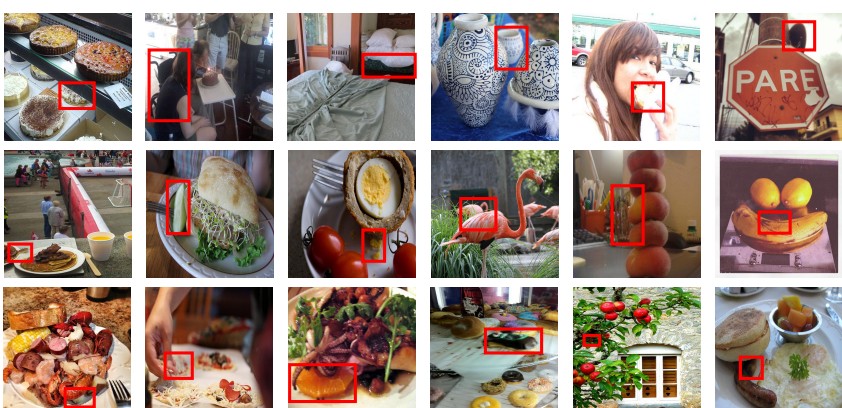

Figure 7: More visualization cases of the low prompt-guided attention targets.

