# OpenReview forum: "VOVTrack: Exploring the Potentiality in Videos for Open-Vocabulary Object Tracking"
_ICLR.cc/2025/Conference — Submitted to ICLR 2025_

### Official Review · Reviewer_h6oh · 2024-10-27

**Soundness:** 2
**Presentation:** 3
**Contribution:** 2
**Rating:** 6
**Confidence:** 4

**Summary:**

This paper proposed a new tracking-related prompt-guided attention for the localization and classification, and develops a self-supervised object similarity learning strategy for the temporal association in OVMOT. Experiments demonstrate that the proposed method achieves state-of-the-art tracking performance compared to previous methods.

**Strengths:**

1. This paper propose a new tracking-related prompt-guided attention for the localization and classification in OVMOT​.
2. The self-supervised learning strategy leverages unlabeled video data for the temporal association, addresses the challenge of training data shortage.
3. Extensive experimental results, demonstrating that VOVTracker outperforms existing methods on TAO benchmark.

**Weaknesses:**

1. Lack of novelty. The self-supervision is common and there are some considerable work in vision tasks. However, this paper just applies it to OVMOT, and does not compare the proposed method with previous methods. What's the main difference and contributions for OVMOT.
2. While VOVtrack leverages unlabeled video data, its performance seems to depend on the quality and representativeness of the videos, which could affect its robustness across different real-world conditions​.
2. The authors should conduct on the ablation sduty of different parameter settings. Such as parameters related to training and inference.

**Questions:**

1. VOVTrack lacks of a clear definition, what's the full name?
2. For Table 1, the propsed VOVtrack performs worse than OVTrack in terms of ClsA metric. What is the reason behind it?
3. For Table 2, it demonstrate the effectiveness of Prompt-guided attention. But the baseline with the self-supervised association module fail to obtain obvious improvements, so that the contribution can not be effectively convinced.

---

> ### Author Response · Authors · 2024-11-22
> **Author Response to Reviewer h6oh [1]**
>
> >1. The self-supervision is common and there are some considerable work in vision tasks. However, this paper just applies it to OVMOT, and does not compare the proposed method with previous methods. What's the main difference and contributions for OVMOT.
>
> We acknowledge that self-supervision approaches are widely applied in vision tasks, which just verifies that self-supervision has a great potential. However, as we know, **the research on self-supervised MOT is scarce[1,2], not to mention the OVMOT.** To the best of our knowledge, this is the first work to apply self-supervision to the OVMOT problem. We next present the main differences and contributions of this work.
>
> First, we clarify the motivation of our method. Compared to MOT, **the self-supervision is more urgently needed for the OVMOT task** because of the lack of available training videos. Previous works, such as OVTrack, use the generated image pairs (data augmentation) for training OVMOT, which ignores the video information for tracking tasks. As discussed in [1], MOT requires associating instances through time, and data augmentation cannot well mimic the occlusions, pose changes, and distortions of real videos. This way, we develop the self-supervised method effectively utilizing consistency in unlabeled raw videos to help the model for association learning.
> The proposed method starts the study of self-supervision for OVMOT, which has a significant contribution to this problem.
>
> Second, we clarify the technique differences of our method. We **specifically design the self-supervision method for OVMOT**. Specifically, we **consider the high diversity of categories and significant inter-frame variations**, which are special in OVMOT tasks but do not appear in previous MOT tasks (focusing on person or vehicle) [1,2].
> This way, we design a category consistency constraint and a long-short interval constraint (see constraint conditions in Eq. 4) to enhance the effectiveness of the self-supervised learning. Moreover, this work considers both the intra- and inter-consistency based on the video continuity but not the image-level similarity (in most previous works) for the self-supervised tracking (see the objective function in Eq. 4). We consider the video continuous similarity for the appearance learning strategy (intra-consistency), as well as the temporal motion feature for the inter-consistency, which are not well leveraged in previous works[1,2].
>
> [1] Segu, Mattia, et al. Walker: self-supervised multiple object tracking by walking on temporal appearance graphs. European Conference on Computer Vision. 2024.
>
> [2] Segu, Mattia, Bernt Schiele, and Fisher Yu. Darth: holistic test-time adaptation for multiple object tracking. Proceedings of the IEEE/CVF International Conference on Computer Vision. 2023.
>
> >2. While VOVTrack leverages unlabeled video data, its performance seems to depend on the quality and representativeness of the videos, which could affect its robustness across different real-world conditions.
>
> Thanks for your comments. To address the robustness concerns regarding our self-supervised learning algorithm, we conducted additional experiments by training our model on the videos in LV-VIS[1] training set (without annotation) instead of the TAO training set.
> We then perform the evaluation on the TAO validation set as in our paper. The experimental results are presented in the table below.
> We can see that our method maintains stable performance even with this dataset shift (from LV-VIS to TAO), achieving very similar results on both Base and Novel metrics, compared to the model trained on TAO. This finding **strongly indicates that our method is dataset-agnostic and exhibits robust generalization capabilities**, as it performs well on the target evaluation set even when trained on a different source dataset.
> #### Table 1: Unsupervised Dataset Fine-Tuning Results Using Various Videos.
> | Method                                |       | **Novel** |       |      |       | **Base** |       |      |
> |---------------------------------------|-------|-----------|-------|------|-------|----------|-------|------|
> |                                       | TETA  | LocA      | AssoA | ClsA | TETA  | LocA     | AssoA | ClsA |
> | OVTrack                 | 27.8  | 48.8       | 33.6  | 1.5  | 35.5  | 49.3      | 36.9  | 20.2 |
> | ours trained on LV-VIS  | 34.0  | 57.9       | 38.5  | 5.7  | 38.0  | 57.9      | 38.6  | 17.5 |
> | ours trained on TAO     | 34.4  | 57.9       | 39.2  | 6.0  | 38.1  | 58.1      | 38.8  | 17.5 |
>
> [1] Wang H, et al. Towards open-vocabulary video instance segmentation." Proceedings of the IEEE/CVF International Conference on Computer Vision. 2023.

---

> ### Author Response · Authors · 2024-11-22
> **Author Response to Reviewer h6oh [2]**
>
> > 3. Insufficient ablation studies with different parameter settings.
>
> Thanks for your suggestions. We considered two representative parameter settings during the training and testing processes and conducted ablation experiments as shown in the below Table.
> During training, we examine the impact of video segment length (in Section 3.2) used for self-supervised training on the final results.
> We can see that our method demonstrates **good robustness to segment length**, performing well across lengths from 8 to 30, with the best results at a length of 24 (used in the experiments).
>
> In the inference process, we evaluate the similarity score threshold (in Section 3.2) used in the association sub-task. We observe a relatively large performance decline when the threshold achieves 0.5. But during the range between 0.3 and 0.45, the Base AssoA remains above 38, and the Novel AssoA stays above 39. This shows that our algorithm is also **not easily influenced by the tracker's similarity threshold setting**.
>
> #### Tabel 2: Ablation studies with different parameter settings.
> | Method                  |       | **Novel**  |       |      |   |       | **Base**  |       |      |
> |-------------------------|-------|------------|-------|------|---|-------|-----------|-------|------|
> |                      |   | TETA  | LocA       | AssoA | ClsA |  TETA  | LocA      | AssoA | ClsA |
> | **Training**         | Segment Length   |       |       |       |       |       |       |       |       |
> |                      | 8                | 37.6  | 57.9  | 37.6  | 17.3  | 33.9  | 57.5  | 38.5  | 5.9   |
> |                      | 16               | 38.0  | 57.7  | 38.1  | 17.5  | 34.1  | 57.2  | 38.9  | 6.1   |
> |                      | **24**           | **38.1** | 58.1  | 38.8  | 17.5  | **34.4** | 57.9  | 39.2  | 6.0   |
> |                      | 30               | 38.0  | 58.1  | 38.4  | 17.6  | 34.1  | 57.5  | 38.9  | 6.0   |
> | **Inference**        | Similarity Threshold |       |       |       |       |       |       |       |       |
> |                      | 0.30             | 38.1  | 58.3  | 38.5  | 17.4  | 34.2  | 58.1  | 39.4  | 5.0   |
> |                      | **0.35**         | **38.1** | 58.1  | 38.8  | 17.5  | **34.4** | 57.9  | 39.2  | 5.9   |
> |                      | 0.40             | 37.8  | 57.7  | 38.1  | 17.5  | 34.1  | 57.8  | 39.0  | 6.0   |
> |                      | 0.45             | 37.8  | 57.8  | 38.0  | 17.5  | 34.1  | 57.8  | 39.0  | 5.9   |
> |                      | 0.50             | 37.4  | 57.4  | 37.6  | 17.4  | 33.8  | 57.4  | 38.8  | 5.3   |
> |                      | 0.55             | 37.0  | 57.0  | 37.0  | 17.0  | 33.5  | 57.0  | 38.5  | 5.3   |
> |                      | 0.60             | 36.6  | 56.7  | 35.3  | 17.8  | 33.0  | 56.9  | 37.1  | 5.1   |
>
> > 4. VOVTrack lacks of a clear definition, what's the full name?
>
> VOVTrack indicates the abbreviation of the paper title, i.e., exploring the potentiality in **V**ideos for **O**pen-**V**ocabulary **Track**ing. We will clarify this in the final revision of our paper.

---

> ### Author Response · Authors · 2024-11-22
> **Author Response to Reviewer h6oh [3]**
>
> >5. For Table 1, the propsed VOVtrack performs worse than OVTrack in terms of ClsA metric. What is the reason behind it?
>
> The baseline method OVTrack **directly uses the publicly available optimal model from DetPro** [1], to obtain the open-vocabulary detection (OVD) results.  Differently, we **re-trained our OVD model** (using the same dataset as [1]) from scratch for OVD **with the proposed tracking-state prompts**.
>
> As shown in the first row of the table below, we find that the pre-trained OVD model used in OVTrack shows a better classification result on the base class, which, however, may have a certain amount of **overfitting**. This is evident from its poor localization and classification performance on the novel class. Differently, our OVD model (third row) significantly improves the novel class classification, with some drops on the base class.
>
> We have to clarify that **performance on the novel class is more important**, not only for the OV tracking problem but also for all OV-related tasks.  As shown in the second row, we can find that the state-of-the-art OV tracker SLAck [2] (ECCV 2024) also shows a performance drop on ClsA of the base class compared to the baseline OVTrack.
>
> Moreover, to further investigate the ClsA performance for the base class. Instead of training from scratch, we **use the OVD model from [1]** as the backbone and **equip it with our proposed method** for evaluation, as shown in the last row in the below table:
>
>   - With the public optimal OVD model, the base class ClsA score improves to **20.0%**, which is comparable with OVTrack (20.2%) and outperforms SLAck (19.1%).
>   - The comprehensive score TETA for the base class achieves the best score of **38.8%**.
>   - However, the ClsA for the novel class becomes 3.2%, although still higher than OVTrack and SLAck, it is obviously **poorer than "Ours"**.
>
> Once again, this work aims to obtain a better performance on the novel class, on which "Ours" maintains the best TETA score of **34.4%**, outperforming OVTrack by a significant margin of **6.6%**.
>
> #### Table 3: Comparison of the impact of using our trained DetPro weights versus public optimal weights.
> | Method                                |       | **Novel** |       |      |       | **Base** |       |      |
> |---------------------------------------|-------|-----------|-------|------|-------|----------|-------|------|
> |                                       | TETA  | LocA      | AssoA | ClsA | TETA  | LocA     | AssoA | ClsA |
> | OVTrack (public OVD model [1])        | 27.8  | 48.8      | 33.6  | 1.5  | 35.5  | 49.3     | 36.9  | **20.2** |
> | SLAck (ECCV 2024 [2])                 | 31.1  | 54.3      | 37.8  | 1.3  | 37.2  | 55.0     | 37.6  | 19.1 |
> | Ours                                  | **34.4** | 5̲7̲.̲9̲ | **39.2** | **6.0** | 3̲8̲.̲1̲ | 5̲8̲.̲1̲ | **38.8** | 17.5 |
> | Ours (public OVD model [1])           | 3̲3̲.̲7̲ | **59.0** | 3̲8̲.̲9̲ | 3̲.̲2̲ | **38.8** | **58.5** | 3̲7̲.̲9̲ | 2̲0̲.̲0̲ |
>
> [1] Du, Yu, et al. Learning to prompt for open-vocabulary object detection with vision-language model. Proceedings of the IEEE/CVF Conference on Computer Vision and Pattern Recognition. 2022.
>
> [2] Li, Siyuan, et al. SLAck: Semantic, Location, and Appearance Aware Open-Vocabulary Tracking. European Conference on Computer Vision. 2024.

---

> ### Author Response · Authors · 2024-11-22
> **Author Response to Reviewer h6oh [4]**
>
> > 6. For Table 2, it demonstrate the effectiveness of Prompt-guided attention. But the baseline with the self-supervised association module fail to obtain obvious improvements, so that the contribution can not be effectively convinced.
>
> We respectfully disagree with the concern regarding the effectiveness of our Self-supervised Learning (SSL) module. We acknowledge that the performance improvement of SSL is smaller than that of prompt-guided attention (Att) under some metrics, which, however, is still obvious. Specifically, as shown in the table below, we calculate the performance improvement of the proposed modules "Att" and "SSL". The row "+ Att" indicates the improvement obtained by the prompt-guided attention, based on which further improvement by SSL is shown in the next row "+ Att + SSL". We can see that, **the performance gains by SSL are substantial**, in the overall TETA metric and the sub-metric of AssoA (association accuracy), which is the self-supervised association module specifically designed for.
>
> We also compared the results of the ablation study of the latest OVMOT method, SLAck (ECCV 2024). It is noteworthy that SLAck conducted ablation experiments *only on the novel class*.
> Here "SLAck w/o ST" refers to the removal of the spatio-temporal fusion module, which is the most important component of SLAck designed for the object association. From the results we can see that, the performance improvement of the main association module ST is smaller than ours, and the final results (including the overall metric TETA and association score AssoA) are also **lower than ours with a relatively large margin.**
>
> #### Table 4: Component Analysis: Effect of Self-Supervised Learning Module.
> | Method                  |       | **Novel**  |       |      |   |       | **Base**  |       |      |
> |:-----------------------|:-----:|:----------:|:-----:|:----:|:-:|:-----:|:---------:|:-----:|:----:|
> |                         | TETA  | LocA       | AssoA | ClsA |   | TETA  | LocA      | AssoA | ClsA |
> | OVTrack                 | 27.8  | 48.8       | 33.6  | 1.5  |   | 35.5  | 49.3      | 36.9  | 20.2 |
> | + Att (Ours w/o SSL)| 31.3 (+3.5) | 55.1  | 34.7 (+1.1) | 4.0 |   | 36.3 (+0.8) | 55.5 | 36.4 (-0.5) | 17.1 |
> | + Att + SSL (Ours)  | **34.4 (+3.1)** | **57.9** | **39.2 (+4.5)** | **6.0** |   | **38.1 (+1.8)** | **58.1** | **38.8 (+1.6)** | 17.5 |
> |                         | **(= 6.6%↑)** |       | **(=5.6%↑)** |       |   | **(=2.6%↑)** |       | **(=1.1%↑)** |       |
> | SLAck w/o ST            | 29.6 (+1.8)  | 53.9       | 34.1  (+0.5)| 0.8  |   | -     | -         | -     | -     |
> | + ST (SLAck)        | 31.1 (+1.5) | 54.3 | 37.8 (+3.7) | 1.3 |   | 37.2  | 55.0  | 37.6  | 19.1  |
> |                         | (= 3.3%↑) |       | (=4.2%↑) |       |   | (=1.7%↑) |       | (=0.7%↑) |       |

---

> ### Author Response · Authors · 2024-11-25
> **Looking forward to further discussions**
>
> Dear Reviewer h6oh,
>
> Thank you for your valuable feedback and for highlighting areas of our work that require further clarification. In our detailed rebuttal, we have thoroughly addressed your concerns regarding the contributions of our self-supervised approach, the generalization capabilities of the proposed method, additional experiments on parameter settings, the explanation for the lower base ClsA results, and evidence demonstrating that our ablation experiments more effectively showcase the method's improvements compared to other SOTA methods.
>
> As the discussion deadline approaches, we would greatly appreciate it if you could review our rebuttal and let us know whether it satisfactorily addresses your concerns.
>
> Thank you once again for your time and effort. We look forward to your response and wish you a wonderful day!
>
> Best regards,
>
> Authors

---

> ### Comment · Reviewer_h6oh · 2024-11-27
> **Feedback to Author Response**
>
> Considering the response addressed part of my concerns, I tend to offer the final rating with 6: marginally above the acceptance threshold.

---

> > ### Author Response · Authors · 2024-11-27
> > **Feedback from authors**
> >
> > Many thanks for your feedback and uprating for this work.

---

### Official Review · Reviewer_n3Po · 2024-10-28

**Soundness:** 2
**Presentation:** 2
**Contribution:** 2
**Rating:** 5
**Confidence:** 5

**Summary:**

This work proposes a series of improvements to the existing OV MOT framework (OVTrack), including a novel prompt-guided attention mechanism and a self-supervised object similarity learning method. With the support of additional video data, these enhancements achieve superior performance on the TAO dataset compared to OVTrack.

**Strengths:**

1. Competitive OV tracking performance on TAO.
2. Self-supervised object similarity learning is compatible with unlabeled video data.

**Weaknesses:**

1. Novelty is limited: The overall architecture of the proposed OVOTrack is based on OVTrack. The specific implementation within the tracking-state aware prompt approach does not exhibit a clear association with the tracking state. Moreover, this method does not assess object quality specifically for tracking scenarios, as it remains overly limited to factors such as occlusion and blurriness. On the other hand, the self-supervised object similarity learning proposed in this work closely resembles QDTrack, with no significant differences observed.
2. Writing requires further improvement: The overall writing is somewhat scattered and lacks coherence, with inconsistencies between the motivation of the paper and the proposed approach, raising concerns of possible over-claim. There is considerable redundancy in language, with some sections being overly simplistic while others are unnecessarily verbose. Additionally, certain assumptions and notations in the methodology are not rigorously standardized (e.g., line 218). Therefore, I believe there is substantial room for improvement in the writing.
3. Experimental setting: The training process described in the paper involves four stages, which makes it overly complex and cumbersome. Compared to OVTrack, this work utilizes a substantial amount of additional TAO video data for training; however, the classification performance on the base categories in the TAO benchmark has declined. Moreover, the ablation study results indicate that the OVOTrack model consistently underperforms in classification metrics, yet no reasonable explanation is provided for this issue. The examples provided in the visualization results are overly simplistic, failing to showcase the model’s true capabilities.

**Questions:**

1. The primary assertion of the paper is that existing OV tracking methodologies do not take into account the states of objects. However, the proposed approaches appear to be largely unrelated to tracking object states, and the introduced prompt, .e.g., occuded, and complete, seems overly naive in my view.
2. The proposed classification method only focuses on the high-quality object. However, exclusively training the classifier on high-quality targets may result in the neglect of low-quality targets that are blurred or occluded. This seems inconsistent with the paper’s claim of addressing issues related to blurring and occlusion in object tracking. Furthermore, could this be the reason for the model’s lower classification performance on the base?

---

> ### Author Response · Authors · 2024-11-22
> **Author Response to Reviewer n3Po [1]**
>
> > 1. Novelty. The specific implementation within the tracking-state aware prompt approach does not exhibit a clear association with the tracking state. Moreover, this method does not assess object quality specifically for tracking scenarios, as it remains overly limited to factors such as occlusion and blurriness.
>
> We first clarify that this method **exhibits a clear association with the tracking state**. Take the “occlusion” as pointed out by the reviewer, for example, it is one of the most difficult challenges in tracking problems, for which many works are particularly designed every year[1][2], to address it. Through prompt learning, we assess the object quality specifically related to the tracking scenarios, and specific states during tracking, such as occlusion/out-of-view/blurriness, etc, to improve the temporal association performance, which can be demonstrated by the results.
>
> We understand the reviewer that the proposed prompt is somewhat simple, which, however, is very effective for the tracking problem. Moreover, we clarify that this is the first work to consider the prompt guided tracking states for OVMOT, even for the MOT. Based on the verification of the effectiveness of this insight, we plan to design more exquisite prompts to further improve the tracking state learning in future work.
>
> Finally, in terms of the novelty of the proposed tracking state prompt, most other reviewers have acknowledged its novelty. For example, Reviewer aVgC pointed out that we develop a *"Novel tracking-related prompt-guided attention"*. Reviewer a19J also presented that it is a *"Novel integration of object states with prompt learning"*. Reviewer h6oh commented that *"Novel tracking-related prompt-guided attention"*.
>
> [1] Cao J, Pang J, Weng X, et al. Observation-centric sort: Rethinking sort for robust multi-object tracking. Proceedings of the IEEE/CVF Conference on Computer Vision and Pattern Recognition. 2023.
>
> [2] Wang Y H, Hsieh J W, Chen P Y, et al. Smiletrack: Similarity learning for occlusion-aware multiple object tracking. Proceedings of the AAAI Conference on Artificial Intelligence. 2024.
>
> > 2. Novelty. The self-supervised object similarity learning proposed in this work closely resembles QDTrack.
>
> The reviewer may have some **misunderstanding** on the QDTrack and the proposed self-supervised method. We have to respectfully argue that our self-supervised similarity learning method is **totally different from that of QDTrack**, which uses the ground truth as supervision.  They are fundamentally different in both underlying principles and implementation details. We then show the detailed reasons.
>
> Above all, regarding input requirements, although automatically generating abundant proposals for training, QDTrack's core mechanism **relies on the ground-truth tracking annotations to compute IoU for positive and negative sample** selection in its quasi-dense similarity learning framework.  In contrast, our approach is **fully annotation-free**, leveraging the cyclic-consistency relationships among multiple frames inherent in the sequences.  This allows our method to excavate the information in the video sequences, while QDTrack only operates on image pairs.
>
> More specifically, the fundamental difference between them is clearly evident in the mathematical formulation as below.
> QDTrack's loss is:  ${L}_{\text{embed}} = \log\left[1 + \sum_k⁺ \sum_k⁻  \exp(\mathbf{v} \cdot \mathbf{k}^- - \mathbf{v} \cdot \mathbf{k}^+)\right]$, which explicitly requires positive target features $\( \mathbf{k}^+ \)$ and $\( \mathbf{k}^- \)$ **derived from ground truth**.  In contrast, our method utilizes transitive-similarity matrix consistency **without any ground truth**, as demonstrated in the following equation:
>
> $ L(\mathbf{E}) = \sum_r \mathrm{ReLU}\left(\max_{c \neq r} \mathbf{E}(r, c) - \mathbf{E}(r, r) + m\right) $,  in which $ \mathbf{E} $ denotes the transitive-similarity matrix that can be automatically calculated from the object features in the video sequences.
>
> >3. Writing requires further improvement.
>
> Thanks for the detailed comments about the writing of this paper. From the comments of other reviewers, such as “Detailed explanation of proposed method” by Reviewer 6vnS, and “Standardized manuscript with fluent writing” by Reviewer aVgC, we could assume that this paper is easy to follow. Certainly, we understand the reviewer may have high standards for writing and point out that there is room for improvement in the writing. As per the reviewer's suggestion, we plan to revise our manuscript by: 1) Provide more details for the sections that are simplistic (in the main paper or the appendix), and simplify others that are verbose. 2) Double-check and improve all the formulations and notations in the methodology to be more standardized.

---

> ### Author Response · Authors · 2024-11-22
> **Author Response to Reviewer n3Po [2]**
>
> >4. Complex training process. The training process described in the paper involves four stages, making it overly complex and cumbersome. Compared to OVTrack, this work utilizes a substantial amount of additional TAO video data for training.
>
> We acknowledge that the training process of this method is more complex compared to OVTrack. However, we clarify that **the model size and running speed are more important factors in practical implementation**. Regarding the model complexity, as shown in the table below, we conducted a comprehensive comparison of computational complexity among our method, the baseline OVTrack (CVPR 2023), and another state-of-the-art comparison approach QDTrack (TPAMI 2023).
>
> We can first see that the model complexity (including the number of parameters and model size) of our method is **comparable with OVTrack and QDTrack**. This is because our method shares the same network architecture with OVTrack.
> Notably, during the inference stage, we optimized the post-processing phase based on OVTrack by removing redundant NMS processing in the RPN and adjusting selection thresholds. These improvements significantly increase the inference speed, achieving an FPS of 15.8, which is about eight times **faster than OVTrack**, while maintaining tracking accuracy. This also results in better FPS performance than QDTrack.
> Overall, considering the inherent complexity of OV tracking tasks, our method achieves competitive performance on computational complexity and running efficiency.
>
> #### Table1: Comparison of computational complexity and efficiency among different tracking methods
> | Method   | Input shape  | Parameters | Model Size | FPS  |
> |----------|--------------|------------|------------|------|
> | QDTrack  | (3,800,1334) | 15.47M     | 298.6M     | 13.8 |
> | OVTrack  | (3,800,1334)  | 16.52M     | 283.77M    | 1.8  |
> | Ours     | (3,800,1334) | 16.52M     | 283.77M    | 15.8 |

---

> ### Author Response · Authors · 2024-11-22
> **Author Response to Reviewer n3Po [3]**
>
> > 5. The classification performance on the base categories in the TAO benchmark has declined. Moreover, the ablation study results indicate that the OVOTrack model consistently underperforms in classification metrics, yet no reasonable explanation is provided for this issue.
>
> We explain for the declined classification performance on base categories. The baseline method OVTrack **directly uses the publicly available optimal model from DetPro** [1], to obtain the open-vocabulary detection (OVD) results.  Differently, we **re-trained our OVD model** (using the same dataset as [1]) from scratch for OVD **with the proposed tracking-state prompts**.
>
> As shown in the first row of the table below, we find that the pre-trained OVD model used in OVTrack shows a better classification result on the base class, which, however, may have a certain amount of **overfitting**. This is evident from its poor localization and classification performance on the novel class. Differently, our OVD model (third row) significantly improves the novel class classification, with some drops on the base class.
>
> We have to clarify that **performance on the novel class is more important**, not only for the OV tracking problem but also for all OV-related tasks.  As shown in the second row, we can find that the state-of-the-art OV tracker SLAck [2] (ECCV 2024) also shows a performance drop on ClsA of the base class compared to the baseline OVTrack.
>
> Moreover, to further investigate the ClsA performance for the base class. Instead of training from scratch, we **use the OVD model from [1]** as the backbone and **equip it with our proposed method** for evaluation, as shown in the last row that:
>   - With the public optimal OVD model, the base class ClsA score improves to **20.0%**, which is comparable with OVTrack (20.2%) and outperforms SLAck (19.1%).
>   - The comprehensive score TETA for the base class achieves the best score of **38.8%**.
>   - However, the ClsA for the novel class becomes 3.2%, although still higher than OVTrack and SLAck, it is obviously **poorer than "Ours"**.
>
> Once again, this work aims to obtain a better performance on the novel class, on which "Ours" maintains the best TETA score of **34.4%**, outperforming OVTrack by a significant margin of **6.6%**.
> #### Table 1: Comparison of the impact of using our trained DetPro weights versus public optimal weights.
> | Method                                |       | **Novel** |       |      |       | **Base** |       |      |
> |---------------------------------------|-------|-----------|-------|------|-------|----------|-------|------|
> |                                       | TETA  | LocA      | AssoA | ClsA | TETA  | LocA     | AssoA | ClsA |
> | OVTrack (public OVD model [1])        | 27.8  | 48.8      | 33.6  | 1.5  | 35.5  | 49.3     | 36.9  | **20.2** |
> | SLAck (ECCV 2024 [2])                 | 31.1  | 54.3      | 37.8  | 1.3  | 37.2  | 55.0     | 37.6  | 19.1 |
> | Ours                                  | **34.4** | 5̲7̲.̲9̲ | **39.2** | **6.0** | 3̲8̲.̲1̲ | 5̲8̲.̲1̲ | **38.8** | 17.5 |
> | Ours (public OVD model [1])           | 3̲3̲.̲7̲ | **59.0** | 3̲8̲.̲9̲ | 3̲.̲2̲ | **38.8** | **58.5** | 3̲7̲.̲9̲ | 2̲0̲.̲0̲ |
>
> [1] Du, Yu, et al. Learning to prompt for open-vocabulary object detection with vision-language model. Proceedings of the IEEE/CVF Conference on Computer Vision and Pattern Recognition. 2022.
>
> [2] Li, Siyuan, et al. SLAck: Semantic, Location, and Appearance Aware Open-Vocabulary Tracking. European Conference on Computer Vision. 2024.

---

> ### Author Response · Authors · 2024-11-22
> **Author Response to Reviewer n3Po [4]**
>
> > 6.  Exclusively training the classifier on high-quality targets may result in the neglect of low-quality targets that are blurred or occluded. This seems inconsistent with the paper’s claim of addressing issues related to blurring and occlusion in object tracking. Furthermore, could this be the reason for the model's lower classification performance on the base.
>
> Thanks for your insightful comments. We know that the training on low-quality targets (hard samples) may be helpful in the classical object detection task.  In the previous closed-set task, where the training set and the test set include consistent classes, it is reasonable to find difficult examples in the training process, which helps to better identify the various targets (of known classes) in the test set.
> However, under the OV setting studied in this work, if **purely pursuing the full even over learning of the base class, the model will not perform well on the novel category**.
> It can be seen from the high base ClsA of OVTrack, but the very low performance of the novel.
>
> The idea of the proposed prompt attention is to filter out high-quality objects for training through tracking states, and then discard low-quality (commonly damaged) training samples so that the network can learn general high-quality ontology representations of the target, rather than mining special representations on the base class of low-quality targets. This is **very important in** the subsequent use of CLIP for **novel class feature alignment**. As can be seen from our experimental results, the tracking performance of our prompt-guided attention training network on novel category is particularly high, far above the baseline, and even higher than the latest SOTA approach published in 2024. This is also the reason for the model's slightly lower classification performance on the base, as pointed out by the reviewer.
>
> Moreover, note that, as discussed in Section 3.2 in the paper,  we apply a multi-stage strategy for sample selection, which does not abandon all the samples with low-quality targets such as blurred or occluded. We **only discard the samples** with attention lower than $d_\rm{low}$, which **are badly damaged and cause disturbance for the training**. We provide such examples in Appendix 6 in the revised paper.
> For the examples with attention scores between $d_\rm{low}$ and $d_\rm{high}$, such as with no serious blur or occlusion, we still maintain them for training.

---

> ### Author Response · Authors · 2024-11-25
> **Looking forward to further discussions**
>
> Dear Reviewer n3Po,
>
> Thank you for your valuable feedback and for highlighting the areas of our work that require further clarification. In our detailed rebuttal, we have thoroughly addressed your concerns regarding the novelty of our method, the differences between our approach and QDTrack, the training process, the explanation for the lower base ClsA results, and the reasons and details behind filtering low-quality objects. Additionally, we have provided more comprehensive visualization results to support our explanations.
>
> As the discussion deadline approaches, we would greatly appreciate it if you could review our rebuttal and let us know whether it satisfactorily addresses your concerns. If so, we also expect you could consider re-evaluating this work.
>
> Thank you once again for your time and effort. We look forward to your response and wish you a wonderful day!
>
> Best regards,
>
> Authors

---

> ### Author Response · Authors · 2024-11-28
> **Looking forward to further discussions with Reviewer n3Po**
>
> Dear Reviewer n3Po,
>
> Thank you again for your valuable comments. We believe we have thoroughly responded to your concerns in our detailed rebuttal. We would appreciate your review of it. Please let us know whether it satisfactorily addresses your concerns. If so, we also expect you could consider re-evaluating this work.
>
> Look forward to your response and wish you a nice day!
>
> Best regards,
>
> Authors

---

> ### Author Response · Authors · 2024-12-02
> **Reminder before the discussion ending**
>
> Dear Reviewer **n3Po**,
>
> With the **approximate ending of the rebuttal**, we would like to know whether our responses satisfactorily address your concerns.
>
> Look forward to your reply and hope you have a nice day!
>
> Best regards,
>
> Authors

---

> > ### Comment · Reviewer_n3Po · 2024-12-03
> >
> > Dear authors,
> >
> > Thank you for addressing my questions. While part of my concerns has been resolved, I still believe that this work represents an incremental improvement upon the existing OVTrack. The tracking state prompt and self-supervised learning design proposed in this paper resemble approaches that have appeared in some earlier works. Maybe this work is more suitable to CVPR or ICCV. At ICLR, as a leading academic conference with the broadest academic influence, I hope to see notable novelty, exceptional insight, solid theory, or comprehensive and robust experiments. Therefore, I believe this work falls below the acceptable standard for ICLR, and I will maintain my current rating.
> >
> > Best,
> > Reviewer n3Po

---

> > > ### Author Response · Authors · 2024-12-03
> > > **Final response to reviewer and AC**
> > >
> > > We thank Reviewer n3Po for reviewing our response and pointing out that the detailed concerns have been addressed. The remaining problem is the matching degree between this paper and ICLR. We respect the stringent specifications of the reviewer for ICLR. We still would like to clarify the contributions of this work in brief, to the reviewer and AC.
> > >
> > > We acknowledge that prompt learning and self-supervised learning have been widely studied before, which, actually, have become the **basic technologies**. Based on them, a bulk of works are published in various conferences, including ICLR. This work has its significant contributions.
> > >
> > > On one hand, we have to clarify that, to the best of our knowledge, **we have not seen any tracking state prompt learning methods in earlier works**.
> > >
> > > On the other hand, the proposed (fully) self-supervised learning strategy makes the model training on videos possible for the OV tracking task, which is **specifically urgently needed since there are still no available (labeled) training videos for this task**. It is also totally different from the earlier works, such as QDTrack (pointed out by the reviewer) requiring supervision.
> > >
> > > No method is created in a vacuum. We do not hope this work to be discounted since it shares some general technologies that appeared in some earlier works but are not similar to specific works.
> > >
> > > We thank the reviewer again for the comments to improve this paper. We also hope the AC considers our above final thoughts into consideration.
> > >
> > > Many thanks.
> > >
> > > Authors

---

### Official Review · Reviewer_a19J · 2024-10-30

**Soundness:** 3
**Presentation:** 3
**Contribution:** 2
**Rating:** 6
**Confidence:** 4

**Summary:**

This work presents a novel open-vocabulary multi-object tracking method that integrates object states based on prompt learning. Different from existing works, it combine OVD and MOT in a unified framework. Some self-supervised losses are designed to learning better object associations. Experiments on public dataset demonstrate the effectiveness of the proposed method.

**Strengths:**

A novel method that integrates object states based on prompt learning is proposed to combine OVD and MOT for open-vocabulary multi-object tracking.

Some self-supervised losses are designed to learning better object associations.

Experiments are conducted on public dataset.

**Weaknesses:**

The following details are unclear. Are the designed prompts only used in training procedure? It would be better if visualized results are provided to validate the effectiveness of these prompts in handling challenging frames with occlusions or motion blur.

There is no comparison with works published in 2024, and the effectiveness of the proposed method is thus not fully validated. The relevent and recent trackers including both closed-set and open-vocabulary ones should be included in comparison.

The results suggest that the proposed method performs abnormal under ClsA metric in both result comparison and ablation study. Though it is better than all methods in most metrics, but it performs worse than OVTrack and OVTrack+RegionCLIP in a clear margin, and also worse than other methods in some cases. The reasons behind these results are not well explained.

Some errors:

Line 256:  into ore framework model

Line 456: Our method

**Questions:**

Some important details and explanations should be provided for clarity.

---

> ### Author Response · Authors · 2024-11-22
> **Author Response to Reviewer a19J [1]**
>
> > 1. Are the designed prompts only used in the training procedure? It would be better if visualized results are provided to validate the effectiveness of these prompts in handling challenging frames with occlusions or motion blur.
>
> Like most CLIP-based prompt tuning methods[1-3], our prompt-guided attention is only implemented during the training phase.
>
> During training, our method effectively obtains and leverages the tracking states. To better illustrate this, we show some visualized results in Figure 3 of the (revised) paper. Specifically, the scores on the images are derived from state-aware prompt attention. We can see that the **targets with high state-aware-attention scores are very distinct**. In contrast, the attention scores for various types (such as occluded, blurred, etc.) of low-score targets are significantly lower. This effectively demonstrates that **the proposed prompt attention can accurately perceive the tracking states of targets**.
>
> Moreover, Figure 4 in the paper visualizes the results on challenging frames with occlusions and motion blur, to validate the effectiveness of our prompts in handling these challenges. Specifically, the first row shows the tracking results from our network trained with prompt-guided attention, while the second row presents the tracking results from OVTrack without using prompt-guided attention. As shown in Figure 4(a), our approach maintains accurate tracking and consistent category identification of a tank even under significant occlusion, showing **substantial improvement over OVTrack**. In Figure 4(b), we present a case of a fast-moving drone with motion blur, where OVTrack misclassifies the target as a "bird", while our method **consistently maintains accurate tracking and correct classification** as a "drone".
>
> [1] Vidit, Vidit, et al. Clip the GAP: A single domain generalization approach for object detection. Proceedings of the IEEE/CVF conference on computer vision and pattern recognition. 2023.
>
> [2] Fahes, Mohammad, et al. Poda: Prompt-driven zero-shot domain adaptation. Proceedings of the IEEE/CVF International Conference on Computer Vision. 2023.
>
> [3] Qi, Daiqing, et al. Generalizing to Unseen Domains via Text-guided Augmentation. European Conference on Computer Vision. 2024.
>
>
> > 2. The proposed method should be compared with the latest methods from 2024, and the effectiveness of the proposed method is thus not fully validated.
>
> As per your suggestion, to fully validate the effectiveness of the proposed method, we compare it with the latest SOTA methods in 2024 for OV tracking, including closed-set tracker MASA[1] (CVPR 2024) and open-vocabulary tracker SLAck[2] (ECCV 2024), in which the ECCV paper is published after the submission of this paper. The evaluation results are shown in the following table.
>
> We can see that our approach **consistently outperforms both SOTA methods** on the comprehensive metric TETA, for both base and novel classes. Notably, our method achieves an improvement of 3.3% over the most recent SOTA method SLAck on TETA of novel class, which is a significant metric specifically for the open-vocabulary tracking task aiming to identify the unseen classes of objects.
>
> #### Table1: Comparison with latest open-vocabulary tracking methods in 2024.
> | Method                                |       | **Novel** |       |      |       | **Base** |       |      |
> |---------------------------------------|-------|-----------|-------|------|-------|----------|-------|------|
> |                                       | TETA  | LocA      | AssoA | ClsA | TETA  | LocA     | AssoA | ClsA |
> | OvTrack (CVPR 2023)   | 27.8 | 48.8 | 33.6  | 1.5  | 35.5 | 49.3 | 36.9  | **20.2** |
> | MASA (CVPR 2024 [1])  | 30.0 | 54.2 | 34.6  | 1.0  | 36.9 | 55.1 | 36.4  | 19.3 |
> | SLAck (ECCV 2024 [2]) | 31.1 | 54.3 | 37.8  | 1.3  | 37.2 | 55.0 | 37.6  | 19.1 |
> | ours                  | **34.4** | **57.9** | **39.2** | **6.0** | **38.1** | **58.1** | **38.8** | 17.5 |
>
>
> [1] Li, Siyuan, et al. Matching Anything by Segmenting Anything. Proceedings of the IEEE/CVF Conference on Computer Vision and Pattern Recognition. 2024.
>
> [2] Li, Siyuan, et al. SLAck: Semantic, Location, and Appearance Aware Open-Vocabulary Tracking. European Conference on Computer Vision. 2024.

---

> ### Author Response · Authors · 2024-11-22
> **Author Response to Reviewer a19J [2]**
>
> > 3. The results suggest that the proposed method performs abnormally under the ClsA metric in both result comparison and ablation study. Though it is better than all methods in most metrics, it performs worse than OVTrack and OVTrack+RegionCLIP by a clear margin, and also worse than other methods in some cases. The reasons behind these results are not well explained.
>
> The baseline method OVTrack **directly uses the publicly available optimal model from DetPro** [1], to obtain the open-vocabulary detection (OVD) results.  Differently, we **re-trained our OVD model** (using the same dataset as [1]) from scratch for OVD **with the proposed tracking-state prompts**.
>
> As shown in the first row of the table below, we find that the pre-trained OVD model used in OVTrack shows a better classification result on the base class, which, however, may have a certain amount of **overfitting**. This is evident from its poor localization and classification performance on the novel class. Differently, our OVD model (third row) significantly improves the novel class classification, with some drops on the base class.
>
> We have to clarify that **performance on the novel class is more important**, not only for the OV tracking problem but also for all OV-related tasks.  As shown in the second row, we can find that the state-of-the-art OV tracker SLAck [2] (ECCV 2024) also shows a performance drop on ClsA of the base class compared to the baseline OVTrack.
>
> Moreover, to further investigate the ClsA performance for the base class. Instead of training from scratch, we **use the OVD model from [1]** as the backbone and **equip it with our proposed method** for evaluation, as shown in the last row that:
>   - With the public optimal OVD model, the base class ClsA score improves to **20.0%**, which is comparable with OVTrack (20.2%) and outperforms SLAck (19.1%).
>   - The comprehensive score TETA for the base class achieves the best score of **38.8%**.
>   - However, the ClsA for the novel class becomes 3.2%, although still higher than OVTrack and SLAck, it is obviously **poorer than "Ours"**.
>
> Once again, this work aims to obtain a better performance on the novel class, on which "Ours" maintains the best TETA score of **34.4%**, outperforming OVTrack by a significant margin of **6.6%**.
> #### Table1: Comparison of the impact of using our trained DetPro weights versus public optimal weights.
> | Method                                |       | **Novel** |       |      |       | **Base** |       |      |
> |---------------------------------------|-------|-----------|-------|------|-------|----------|-------|------|
> |                                       | TETA  | LocA      | AssoA | ClsA | TETA  | LocA     | AssoA | ClsA |
> | OVTrack (public OVD model [1])        | 27.8  | 48.8      | 33.6  | 1.5  | 35.5  | 49.3     | 36.9  | **20.2** |
> | SLAck (ECCV 2024 [2])                 | 31.1  | 54.3      | 37.8  | 1.3  | 37.2  | 55.0     | 37.6  | 19.1 |
> | Ours                                  | **34.4** | 5̲7̲.̲9̲ | **39.2** | **6.0** | 3̲8̲.̲1̲ | 5̲8̲.̲1̲ | **38.8** | 17.5 |
> | Ours (public OVD model [1])           | 3̲3̲.̲7̲ | **59.0** | 3̲8̲.̲9̲ | 3̲.̲2̲ | **38.8** | **58.5** | 3̲7̲.̲9̲ | 2̲0̲.̲0̲ |
>
> [1] Du, Yu, et al. Learning to prompt for open-vocabulary object detection with vision-language model. Proceedings of the IEEE/CVF Conference on Computer Vision and Pattern Recognition. 2022.
>
> [2] Li, Siyuan, et al. SLAck: Semantic, Location, and Appearance Aware Open-Vocabulary Tracking. European Conference on Computer Vision. 2024.
>
> &nbsp;
>
> Thanks for your valuable comments about the unclear details and SOTA comparison, which are helpful in improving this paper. We are happy to add them to the revised version. If you have any further questions, we are happy to discuss them with you. If we have resolved your concerns, we sincerely hope you might consider increasing the score. Thank you again.

---

> ### Author Response · Authors · 2024-11-25
> **Sincere Thanks for Your Support and Feedback**
>
> Dear Reviewer a19J,
>
> Thank you for your valuable feedback. We are pleased to have addressed your concerns and greatly appreciate your support in increasing the score.
>
> Best regards,
>
> Authors

---

### Official Review · Reviewer_aVgC · 2024-11-02

**Soundness:** 3
**Presentation:** 3
**Contribution:** 3
**Rating:** 6
**Confidence:** 4

**Summary:**

The paper introduces Open-vocabulary multi-object tracking (OVMOT), a significant challenge that involves detecting and tracking various object categories in videos, including both known (base classes) and unknown (novel classes) categories. The authors critique existing OVMOT methods for treating open-vocabulary object detection (OVD) and multi-object tracking (MOT) as separate modules, primarily focusing on image-based approaches. To address this, they present VOVTrack, which integrates object states relevant to MOT with video-centric training, approaching the challenge from a video object tracking perspective. VOVTrack features a prompt-guided attention mechanism that enhances the localization and classification of dynamic objects, and it employs a self-supervised object similarity learning technique for tracking using raw, unlabeled video data. Experimental results demonstrate that VOVTrack outperforms current methods, establishing it as a leading solution for open-vocabulary tracking tasks.

**Strengths:**

1. This manuscript is standardized and the writing is fluent, and the content is easy to understand.

2. This manuscript proposes a new tracking-related prompt-guided attention for the localization and classification (detection) in the open vocabulary tracking problem. This method takes notice of the states of the time-varying objects during tracking, which is different from the open-
vocabulary object detection from a single image.

3. This manuscript develops a self-supervised object similarity learning strategy for the temporal association (tracking) in the OVMOT problem. This strategy, for the first time, makes full use of the raw video data without annotation for OVMOT training, thus addressing the problem of training data shortage and eliminating the heavy burden of annotation of OVMOT.

4. Experimental results on the public benchmark demonstrate that the proposed VOVTrack achieves the best performance with the same training dataset (annotations), and comparable performance with the methods using a large dataset (CC3M) for training.

**Weaknesses:**

1. The experimental tables lack details on model complexity. It would be helpful to include a table or section comparing FLOPs, parameters, model size, and FPS across the different methods evaluated, including the baseline OVTrack method and other state-of-the-art approaches.

**Questions:**

Please refer to the concerns and issues raised in the "Weaknesses".

---

> ### Author Response · Authors · 2024-11-22
> **Author Response to Reviewer aVgC**
>
> > 1. The experimental tables lack details on model complexity. It would be helpful to include a table or section comparing FLOPs, parameters, model size, and FPS across the different methods evaluated, including the baseline OVTrack method and other state-of-the-art approaches.
>
> Firstly, we thank you for your valuable feedback and recognition of our work. Regarding the model complexity, as shown in the table below, we conducted a comprehensive comparison of computational complexity among our method, the baseline OVTrack (CVPR 2023), and another state-of-the-art comparison approach QDTrack (TPAMI 2023). We can first see that the model complexity (including the FLOPs, number of parameters and model size) of our method is comparable with OVTrack and QDTrack. This is because our method shares the same network architecture with OVTrack.
>
> Notably, during the inference stage, we optimized the post-processing phase based on OVTrack by removing redundant NMS processing in the RPN and adjusting selection thresholds. These improvements significantly increase the inference speed, achieving an FPS of 15.8, which is about **eight times higher than OVTrack**, while maintaining tracking accuracy. This also results in **better FPS performance compared to QDTrack**.
>
> Overall, considering the inherent complexity of OV tracking tasks, our method achieves competitive performance on computational complexity and running efficiency.
>
> | Method   | Input shape  | Test FLOPs | Train FLOPs | Parameters | Model Size | FPS  |
> |----------|--------------|------------|-------------|------------|------------|------|
> | QDTrack  | (3,800,1334) | 398.93G    | 401.17G     | 15.47M     | 298.6M     | 13.8 |
> | OVTrack  | (3,800,1334) | 423.60G    | 410.72G     | 16.52M     | 283.77M    | 1.8  |
> | Ours     | (3,800,1334) | 423.60G    | 413.19G     | 16.52M     | 283.77M    | 15.8 |
>
> We will include this comparison and analysis in the final versions of the paper (or appendix). Once again, we sincerely thank you for recognizing the novelty of this work. If we have adequately addressed your concerns, we genuinely hope you might consider increasing the score, as it is very important to us.

---

> ### Author Response · Authors · 2024-11-25
> **Looking forward to further discussions**
>
> Dear Reviewer aVgC,
>
> Thank you for your recognition and support of our work. In our detailed rebuttal, we have thoroughly addressed your concerns regarding the model's complexity and provided comprehensive statistical analyses.
>
> As the discussion deadline approaches, we would greatly appreciate it if you could review our rebuttal and let us know whether it satisfactorily addresses your concerns.  If so, we also genuinely hope you might consider increasing the rating.
>
> Thank you once again for your time and effort. We look forward to your response and wish you a wonderful day!
>
> Best regards,
> Authors

---

### Official Review · Reviewer_6vnS · 2024-11-08

**Soundness:** 3
**Presentation:** 2
**Contribution:** 2
**Rating:** 5
**Confidence:** 5

**Summary:**

The paper introduces Tracking-state-aware prompt guided attention, enabling the network to learn the detection of objects in different tracking states. A self-supervised approach is adopted to train tracking, leveraging large-scale, unlabeled video data across various categories. The experimental results on TAO datasets indicate that the proposed method achieves advanced performance.

**Strengths:**

1. The paper offers a detailed explanation of the proposed method. Utilizing unlabeled data for self-supervision serves, to some extent, as an alternative to address the current scarcity of large vocabulary tracking data.
2. The results show good performance in comparisons on the OVMOT benchmark.

**Weaknesses:**

1. The CIsA for basic categories in your method is lower than that of the baseline method OVTrack. Given that basic categories account for the majority of targets, the decrease on CIsA appears to reflect more than just typical fluctuation effects. Could this imply that your model's approach introduces a degree of conflict between tracking and classification based on the baseline?
2. There is a lack of visualization and analysis for the prompt guided attention, which does not adequately demonstrate its direct impact.

**Questions:**

1. Considering CLIP's training process, is there a significant distinction in the representations output by CLIP when tracking-state-aware prompts, such as 'unoccluded' and 'occluded', are provided?
2. Why does your method improve the ClA for novel categories compared to the baseline OVTrack but decrease scores for base categories? Does this suggest that your model introduces a conflict between tracking and classification?
3. Could you include a visual representation and analysis of the prompt guided attention? This would more intuitively demonstrate its role and effect.

---

> ### Author Response · Authors · 2024-11-22
> **Author Response to Reviewer 6vnS [1]**
>
> >1. The CIsA for basic categories in your method is lower than that of the baseline method OVTrack.
>
> The baseline method OVTrack **directly uses the publicly available optimal model from DetPro** [1], to obtain the open-vocabulary detection (OVD) results.  Differently, we **re-trained our OVD model** (using the same dataset as [1]) from scratch for OVD **with the proposed tracking-state prompts**.
>
> As shown in the first row of the table below, we find that the pre-trained OVD model used in OVTrack shows a better classification result on the base class, which, however, may have a certain amount of **overfitting**. This is evident from its poor localization and classification performance on the novel class. Differently, our OVD model (third row) significantly improves the novel class classification, with some drops on the base class.
>
> We have to clarify that **performance on the novel class is more important**, not only for the OV tracking problem but also for all OV-related tasks.  As shown in the second row, we can find that the state-of-the-art OV tracker SLAck [2] (ECCV 2024) also shows a performance drop on ClsA of the base class compared to the baseline OVTrack.
>
> Moreover, to further investigate the ClsA performance for the base class. Instead of training from scratch, we **use the OVD model from [1]** as the backbone and **equip it with our proposed method** for evaluation, as shown in the last row in Table R1 that:
>   - With the public optimal OVD model, the base class ClsA score improves to **20.0%**, which is comparable with OVTrack (20.2%) and outperforms SLAck (19.1%).
>   - The comprehensive score TETA for the base class achieves the best score of **38.8%**.
>   - However, the ClsA for the novel class becomes 3.2%, although still higher than OVTrack and SLAck, it is obviously **poorer than "Ours"**.
>
> Once again, this work aims to obtain a better performance on the novel class, on which "Ours" maintains the best TETA score of **34.4%**, outperforming OVTrack by a significant margin of **6.6%**.
> #### Table R1: Comparison of the impact of using our trained DetPro weights versus public optimal weights.
> | Method                                |       | **Novel** |       |      |       | **Base** |       |      |
> |---------------------------------------|-------|-----------|-------|------|-------|----------|-------|------|
> |                                       | TETA  | LocA      | AssoA | ClsA | TETA  | LocA     | AssoA | ClsA |
> | OVTrack (public OVD model [1])        | 27.8  | 48.8      | 33.6  | 1.5  | 35.5  | 49.3     | 36.9  | **20.2** |
> | SLAck (ECCV 2024 [2])                 | 31.1  | 54.3      | 37.8  | 1.3  | 37.2  | 55.0     | 37.6  | 19.1 |
> | Ours                                  | **34.4** | 5̲7̲.̲9̲ | **39.2** | **6.0** | 3̲8̲.̲1̲ | 5̲8̲.̲1̲ | **38.8** | 17.5 |
> | Ours (public OVD model [1])           | 3̲3̲.̲7̲ | **59.0** | 3̲8̲.̲9̲ | 3̲.̲2̲ | **38.8** | **58.5** | 3̲7̲.̲9̲ | 2̲0̲.̲0̲ |
>
> [1] Du, Yu, et al. Learning to prompt for open-vocabulary object detection with vision-language model. Proceedings of the IEEE/CVF Conference on Computer Vision and Pattern Recognition. 2022.
>
> [2] Li, Siyuan, et al. SLAck: Semantic, Location, and Appearance Aware Open-Vocabulary Tracking. European Conference on Computer Vision. 2024.
>
> > 2. Why does your method improve the ClA for novel categories compared to the baseline OVTrack but decrease scores for base categories? Does this suggest that your model introduces a conflict between tracking and classification?
>
> We further investigate whether our model introduces a conflict between tracking and classification tasks. Specifically, we evaluate the classification performance of the detection results frame by frame (without considering tracking) and compare it to the performance when tracking is integrated into our method.
>
> As shown in the Table 2 below, our method **not only maintains but also improves the classification performance, achieving a 1.7\% improvement on base categories and a 1.2\% increase on novel categories**. These results validate that, as discussed in the manuscript, our model **introduces not a conflict, but a mutual assist mechanism between tracking and classification tasks**.
>
> #### Table R2: Comparison of classification results with and without the tracking assist.
>  | Method     | Base ClsA | Novel ClsA |
>  |------------|-----------|------------|
>  | only with Detection  | 15.8      | 4.8        |
>  | + with Tracking   | 17.5 (**+1.7**) | 6.0 (**+1.2**) |

---

> ### Author Response · Authors · 2024-11-22
> **Author Response Reviewer 6vnS [2]**
>
> > 3. Considering CLIP's training process, does CLIP significantly differentiate between various tracking-state-aware prompts?
>
> Thanks for your thoughtful comment. We clarify that CLIP's training process utilizes 400 million image-text pairs for pre-training, utilizing complete textual descriptions rather than pure object classes for language embedding. Note that, such raw textual descriptions actually provide comprehensive coverage of various object states, which can be valuable for the downstream tasks.
>
> We acknowledge that most existing works aim to leverage the CLIP model to obtain the prior between object patch and class prompt.
> Note that, some recent studies have begun to **extract more implicit information from CLIP**, such as recent works have discovered that CLIP can even discern the aesthetic attributes of images[1,2], which are more implicit to be discriminative.
>
> This way, in this work, we aim to make full use of CLIP to differentiate and leverage the various tracking-state-aware prompts for the OVMOT task.
>
> [1] Sheng, Xiangfei, et al. Aesclip: Multi-attribute contrastive learning for image aesthetics assessment. ACM International Conference on Multimedia, 2023.
>
> [2] Huang, Yipo, et al. Multi-Modality Multi-Attribute Contrastive Pre-Training for Image Aesthetics Computing. IEEE Transactions on Pattern Analysis and Machine Intelligence. 2024.
>
> >4. There is a lack of visualization and analysis for the prompt-guided attention, which does not adequately demonstrate its direct impact. Could you include a visual representation and analysis of the prompt-guided attention? This would more intuitively demonstrate its role and effect.
>
> We have presented the visualization to show the impact of prompt attention in Figure 3 (in the paper). Based on your suggestion, we have updated it in the revised paper to more intuitively demonstrate the direct effects of prompt attention.
>
> Specifically, the scores on the images are derived from state-aware prompt attention. It is evident that **the targets with high state-aware-attention scores are very distinct**. In contrast, the attention scores for various types (incomplete, occluded, etc.) of low-score targets are significantly lower. This effectively demonstrates that the proposed prompt attention can **accurately perceive the tracking states of targets**.
>
>   &nbsp;
>
> If you have any further questions, we would be happy to address them and discuss them with you. Otherwise, we sincerely hope you might consider increasing the score, which will allow us the opportunity to incorporate your valuable suggestions into the camera-ready version.

---

> > ### Comment · Reviewer_6vnS · 2024-12-03
> >
> > I appreciate the authors' comprehensive responses which have adddressed some of  my concerns.   But I still not satisfied with
> > this explanation that the basic categories  is lower than that of the baseline method OVTrack.  Anyway, the decision is left to AC.

---

> > > ### Author Response · Authors · 2024-12-03
> > > **Final response to reviewer and AC**
> > >
> > > We thank Reviewer 6vnS for reviewing our response and pointing out that the previous concerns have been addressed except for explaining the lower classification of base categories. Finally, we would like to provide a concise explanation to the reviewer and AC as factors for evaluation.
> > >
> > > 1) The OV tracking problem aims to simultaneously address three sub-tasks of object localization, association, and classification (using LocA, AssoA, and ClsA metrics) for both base and novel categories. Classification is only for one of the sub-tasks, **our method performs very well on the comprehensive metrics TETA (for novel and base categories)**. Besides, as clarified in many previous OV problem papers, **the evaluation of novel categories is more important**, where our method improves the classification significantly.
> > >
> > > 2) The reason for the lower classification for base categories is that we do not directly apply the pre-trained OVD model (used in OVTrack), which is overfitted on the base categories. To verify this, we have further provided the degraded version using the pre-trained OVD model (in the last row of Table R1), which **outperforms the state-of-the-art (homochronous) OV tracking method SLAck (published in ECCV 2024) for all metrics**.
> > >
> > > In summary, we believe the proposed method maintains a leading performance edge at the stage of being. For the burgeoning OV tracking problem, we aim to explore the significant improvement of overall performance, especially for the newly involved (compared to the classical close-set problem) and more challenging novel categories.
> > >
> > > As presented by the reviewer, the decision is left to AC. We just hope the AC can consider our final statement above in the final decision.
> > >
> > > Many thanks.
> > >
> > > Authors

---

> > > ### Comment · Reviewer_6vnS · 2024-12-03
> > >
> > > Thank you to the author for the effort put into the rebuttal. After seeing the final response to the reviewer and AC, I noticed there is a slight misunderstanding regarding our concerns, and I would like to clarify the points of dissatisfaction.
> > >
> > > 1. Distinction in the representations output by CLIP when using tracking-state-aware prompts
> > >
> > > The author’s reply remains insufficiently persuasive. Whether the semantic difference between "unoccluded" and "occluded" leads to embedding differences in CLIP’s outputs requires experimental validation.
> > >
> > > Furthermore, the example provided by the author appears flawed.
> > >
> > > For instance, Aesclip is an enhanced version of CLIP with specialized training, enabling it to assess aesthetics. However, as the author is using the original version of CLIP, whether it can effectively distinguish between the semantics of "unoccluded" and "occluded" remains unverified. This raises questions about the true cause of the performance improvement.
> > >
> > > [1] Sheng, Xiangfei, et al. Aesclip: Multi-attribute contrastive learning for image aesthetics assessment. ACM International Conference on Multimedia, 2023.
> > >
> > > 2. Potential conflict between tracking and classification
> > >
> > > According to the author’s paper, the proposed method, VOVTrack, is introduced as an improvement based on OVTrack. It is important to examine how introducing this method affects both classification and tracking performance.
> > >
> > > I appreciate the author for providing the comparison experiment results in the rebuttal.
> > >
> > > However, the rebuttal experiments focus on whether adding a tracking structure impacts classification performance, rather than the effects brought by the improvements in VOVTrack.
> > >
> > > Additionally, from the ablation table, it appears that under all settings, the introduction of the author’s method significantly reduces the classification performance on base classes, which raises concerns about its reasonableness.
> > >
> > > I acknowledge the author for providing visualization results regarding the prompt-guided attention.
> > >
> > > I appreciate the author’s rebuttal, but for the reasons above, I maintain my Rating: 5.

---

> ### Author Response · Authors · 2024-11-25
> **Looking forward to further discussions**
>
> Dear Reviewer 6vnS,
>
> Thank you for your insightful feedback on our paper. We have carefully addressed each of your comments in our detailed rebuttal, including explanations for the lower base ClsA, the interplay between classification and tracking, the effectiveness of CLIP in state awareness, and enhanced visualizations of prompt-guided attention.
>
> As the discussion deadline approaches, we would greatly appreciate it if you could review our rebuttals and let us know whether they satisfactorily address your concerns.
>
> Thank you once again for your time and effort. We look forward to your response and hope you have a wonderful day!
>
> Best regards,
>
> Authors

---

> ### Author Response · Authors · 2024-11-28
> **Looking forward to further discussions with Reviewer 6vnS**
>
> Dear Reviewer 6vnS,
>
> Thank you again for your effort and insightful comments. We have responded to all your comments in detail in the rebuttal. We would greatly appreciate your review of our rebuttals. Please let us know whether they satisfactorily address your concerns.
>
> We look forward to your response and hope you have a nice day!
>
> Best regards,
>
> Authors

---

> ### Author Response · Authors · 2024-12-02
> **Reminder before the discussion ending**
>
> Dear Reviewer **6vnS**,
>
> With the **approximate ending of the rebuttal**, we would like to know whether our responses satisfactorily address your concerns.
>
> Look forward to your reply and hope you have a nice day!
>
> Best regards,
>
> Authors

---

> ### Author Response · Authors · 2024-12-03
> **Further Response**
>
> Since the feedback is so late, we can only provide a quick response.
>
> For Q1, we can see from Figure 3 (in the paper) that, although with the original CLIP, the attention scores for various types (occluded, incomplete, etc.) of low-score targets are significantly lower. The **visual results effectively demonstrate that the proposed prompt attention can accurately perceive the tracking states of targets**.
>
> For Q2, VOVTrack is based on OVTrack. As shown in the main results, compared to OVTrack, introducing our method only decreases the classification performance and improves all other metrics significantly. Specifically, **for novel classes, the classification and tracking are both improved**. There is no conflict to be observed between tracking and classification. The decrease in base class classification has been explained in detail before.
>
> Thanks.

---

### Author Response · Authors · 2024-12-02
**Summary of rebuttal**

Dear all reviewers and ACs,

With the approximate ending of the rebuttal period, we would like to make a summary of this work.

This work received the initial ratings of 5 (6vnS), 6 (aVgC), 5 (a19J), 5 (n3Po), 5 (h6oh). After the rebuttal, Reviewers a19J and h6oh have replied and both raised the rating to 6, making the current rating as 5 (6vnS), 6 (aVgC), 6 (a19J), 5 (n3Po), 6 (h6oh).

For Reviewer 6vnS, we have thoroughly addressed all the comments in the rebuttal. In particular, including the explanations for the lower base ClsA (also asked by reviewers a19J and h6oh and have been addressed), the positive interplay between classification and tracking (verified by the new experiments), the effectiveness of CLIP in tracking state awareness (explained and supported by public works), and improved visualizations of prompt-guided attention (in the revised paper).

For Reviewer n3Po, the comments have also been thoroughly addressed. 1) About the novelty. For the proposed tracking-specific state prompt, most other reviewers have acknowledged its novelty, including ‘Novel tracking-related prompt-guided attention’ (Reviewer aVgC), ‘Novel integration of object states with prompt learning’ (Reviewer a19J) and ‘Novel tracking-related prompt-guided attention' (Reviewer h6oh). For the self-supervised learning strategy, we have clarified that it is a fully self-supervised method totally different from QDTrack (actually a supervised one) pointed out by Reviewer n3Po. Its novelty has also been approved by Reviewer h6oh after rebuttal. 2) For the writing, this paper received a commendation from Reviewer 6vnS ('Detailed explanation of proposed method'), and Reviewer aVgC ('Standardized manuscript with fluent writing'). We also promise to further improve it as per Reviewer n3Po's suggestions. 3) For the experiments, we have also added the results and analysis in the detailed response, which will be included in our camera-ready version.

At last, please allow us to highlight again the novelty and contribution of this work. First, based on the recently popular prompt learning, this work estimates the **object states specifically for tracking**, such as occlusion/out-of-view/blurriness, etc, through prompt learning, which is **non-trivial and highly helpful for the open-vocabulary (OV) tracking, especially the (more important) novel class** (see the response to Reviewer h6oh [4]). Moreover, this is the first work to apply the self-supervision for OV tracking task, which **makes the model training on videos possible**. This is **urgently needed for the OV tracking** task since there are still no available (labeled) training videos. Benefitting from these contributions, the proposed method outperforms the homochronous works published in 2024 (see the response to Reviewer a19J [1]).

We sincerely hope the reviewers and ACs can take the above points into consideration in the final decision.

Many thanks.

Authors

---

### Meta-Review · Area_Chair_bETW · 2024-12-20

**Metareview:**

Summary:
This paper proposes OVTracker that integrates object states relevant to MOT and video-centric training for the open vocabulary tracking task. A prompt-guided attention mechanism is developed for more accurate localization and classification and a self-supervised object similarity learning technique is formulated for enhancing object association. The experimental results on the OVMOT dataset are given.

The main strengths are 1）strong performance, and 2) well-structured and easy to follow.
The main weaknesses are 1) limited novelty (the proposed method is an incremental work of VTrack) and 2) More experiments and visualization are needed.

After the discussion phase, the main issue of lack of novelty still remains, as recognized by Reviewers 6vnS and n3Po. As these issues significantly degrade the contribution of this paper, the AC does not recommend it be accepted at this conference.
The authors are encouraged to take the comments into consideration for their future submissions.

**Additional Comments On Reviewer Discussion:**

This work received initial ratings of 5 (6vnS), 6 (aVgC), 5 (a19J), 5 (n3Po), and 5 (h6oh). During the rebuttal, the authors provided more experimental results and clarifications addressing the issues related to ablation studies and writing.

After the rebuttal, Reviewers a19J and h6oh raised the rating to 6.
However, Reviewers 6vnS and n3Po highlighted that the issues of limited novelty and technique contribution remain.
The final ratings are 66655.

---

### Decision · Program_Chairs · 2025-01-22

Reject